# Deciphering Medulloblastoma: Epigenetic and Metabolic Changes Driving Tumorigenesis and Treatment Outcomes

**DOI:** 10.3390/biomedicines13081898

**Published:** 2025-08-04

**Authors:** Jenny Bonifacio-Mundaca, Sandro Casavilca-Zambrano, Christophe Desterke, Íñigo Casafont, Jorge Mata-Garrido

**Affiliations:** 1National Tumor Bank, Department of Pathology, National Institute of Neoplastic Diseases, Surquillo 15038, Peru; jenny.bonifacio@upch.pe (J.B.-M.); scasavilca@inen.sld.pe (S.C.-Z.); 2Faculté de Médecine du Kremlin Bicêtre, Université Paris-Saclay, 94270 Le Kremlin-Bicêtre, France; christophe.desterke@inserm.fr; 3Anatomy & Cell Biology Department, School of Medicine, University of Cantabria, 39011 Santander, Spain; inigo.casafont@unican.es; 4The Nanomedicine Group, Institute Valdecilla-IDIVAL, 39011 Santander, Spain

**Keywords:** medulloblastoma, pediatric cancer, epigenetics, metabolism

## Abstract

Background/Objectives: Medulloblastoma is the most common malignant brain tumor in children and comprises four molecular subtypes—WNT, SHH, Group 3, and Group 4—each with distinct genetic, epigenetic, and metabolic features. Increasing evidence highlights the critical role of metabolic reprogramming and epigenetic alterations in driving tumor progression, therapy resistance, and clinical outcomes. This review aims to explore the interplay between metabolic and epigenetic mechanisms in medulloblastoma, with a focus on their functional roles and therapeutic implications. Methods: A comprehensive literature review was conducted using PubMed and relevant databases, focusing on recent studies examining metabolic pathways and epigenetic regulation in medulloblastoma subtypes. Particular attention was given to experimental findings from in vitro and in vivo models, as well as emerging preclinical therapeutic strategies targeting these pathways. Results: Medulloblastoma exhibits metabolic adaptations such as increased glycolysis, lipid biosynthesis, and altered amino acid metabolism. These changes support rapid cell proliferation and interact with the tumor microenvironment. Concurrently, epigenetic mechanisms—including DNA methylation, histone modification, chromatin remodeling, and non-coding RNA regulation—contribute to tumor aggressiveness and treatment resistance. Notably, metabolic intermediates often serve as cofactors for epigenetic enzymes, creating feedback loops that reinforce oncogenic states. Preclinical studies suggest that targeting metabolic vulnerabilities or epigenetic regulators—and particularly their combination—can suppress tumor growth and overcome resistance mechanisms. Conclusions: The metabolic–epigenetic crosstalk in medulloblastoma represents a promising area for therapeutic innovation. Understanding subtype-specific dependencies and integrating biomarkers for patient stratification could facilitate the development of precision medicine approaches that improve outcomes and reduce long-term treatment-related toxicity in pediatric patients.

## 1. Introduction

Medulloblastoma is the most common malignant brain tumor in children and the leading cause of cancer-related mortality in pediatric patients [1]. Arising in the cerebellum, this tumor represents about 20% of all central nervous system (CNS) cancers in children and has a peak incidence between ages three and eight [2]. Despite extensive research and treatment improvements, medulloblastoma remains a highly aggressive tumor with significant treatment challenges. Current therapies typically combine surgery, craniospinal radiation, and chemotherapy, but these carry long-term side effects that can profoundly impact quality of life, particularly for pediatric patients whose developing CNS is vulnerable to radiation and cytotoxic treatments [2]. Thus, while survival rates have improved to approximately 70–80% for certain subgroups, the risk of recurrence and severe treatment-related complications remains a formidable hurdle.

In recent years, the understanding of medulloblastoma has evolved from a view of it as a singular disease to recognizing it as a collection of biologically distinct subtypes. Based on molecular and genetic profiling, medulloblastoma is currently classified into four primary subgroups: WNT, SHH (Sonic Hedgehog), Group 3, and Group 4 [3]. These subgroups are defined by unique molecular characteristics, genetic mutations, and clinical behaviors, which have prognostic and therapeutic implications. For instance, WNT-subgroup medulloblastomas tend to have a favorable prognosis with a high survival rate [4], while Group 3 tumors, frequently associated with MYC amplification, often exhibit aggressive growth and poorer outcomes [5]. Group 4 tumors, the most common subtype, exhibit variable prognosis, and SHH-subgroup tumors are often associated with genetic mutations in the SHH signaling pathway and tend to have an intermediate prognosis [4]. This molecular diversity underscores the need for subgroup-specific treatments that can effectively target the distinct biological drivers of each subtype.

Recent advances in cancer biology have highlighted the critical role of metabolic and epigenetic pathways in supporting tumor growth and survival, and medulloblastoma is no exception [6]. Tumor cells are known to rewire their metabolism to fuel rapid cell division, evade apoptosis, and adapt to hypoxic or nutrient-poor environments—a phenomenon known as metabolic reprogramming. This process often involves increased glycolysis (even in the presence of oxygen, known as the Warburg effect), alterations in oxidative phosphorylation, lipid synthesis, and amino acid metabolism [7]. Medulloblastoma tumors, particularly the SHH and Group 3 subgroups, show distinct metabolic profiles that facilitate their aggressive growth and resistance to conventional therapies [8]. Understanding these metabolic pathways is essential, as they represent potential therapeutic targets. For example, inhibitors of glycolysis, fatty acid synthesis, or glutamine metabolism could selectively disrupt the metabolic needs of medulloblastoma cells, curbing their growth and improving patient outcomes.

In parallel, epigenetic mechanisms—including DNA methylation, histone modification, and chromatin remodeling—play a pivotal role in medulloblastoma by regulating gene expression without altering the underlying DNA sequence [9]. Each medulloblastoma subgroup is associated with a specific pattern of epigenetic alterations that influence tumor behavior, differentiation, and response to therapy [10]. For instance, aberrant DNA methylation and histone modifications are known to contribute to the maintenance of the stem-like properties in tumor cells, allowing them to proliferate and evade differentiation signals [11]. Additionally, non-coding RNAs, such as microRNAs and long non-coding RNAs, are increasingly recognized as key epigenetic regulators of medulloblastoma biology, influencing cell cycle progression, apoptosis, and metastasis [12]. Targeting these epigenetic regulators holds promise, as epigenetic changes are reversible and could be modulated to halt tumor growth or enhance sensitivity to existing therapies.

The interaction between metabolic and epigenetic pathways adds an additional layer of complexity and potential therapeutic opportunity. Metabolites generated through cellular metabolism, such as acetyl-CoA, α-ketoglutarate, and S-adenosylmethionine (SAM), act as cofactors or substrates for epigenetic enzymes, thereby linking metabolic states with gene expression. In medulloblastoma, this crosstalk suggests that disrupting either the metabolic or epigenetic axis may lead to therapeutic vulnerabilities, especially if combined with other targeted treatments.

The primary goal of this review is to examine the role of metabolic and epigenetic pathways in medulloblastoma and to explore how these processes contribute to tumor development, maintenance, and therapeutic resistance. This review will cover the latest findings in metabolic reprogramming across medulloblastoma subgroups, identifying key metabolic alterations and their implications for tumor cell proliferation, survival, and metastasis. Additionally, we will explore the distinct epigenetic landscapes of each medulloblastoma subgroup, focusing on how DNA methylation patterns, histone modifications, and chromatin structure impact gene expression and drive tumor behavior.

## 2. Medulloblastoma Subtypes and Molecular Classification

### 2.1. Current Classification

Medulloblastoma is no longer viewed as a single disease entity but rather as a collection of distinct molecular subtypes, each with unique genetic, epigenetic, and clinical features [1,2,3,4,5]. Comprehensive genomic and transcriptomic studies have classified medulloblastoma into four primary subtypes: WNT, SHH, Group 3, and Group 4 [13]. This classification reflects the complexity and heterogeneity of the disease, with each subtype differing in terms of molecular drivers, histopathology, patient prognosis, and response to therapy (Figure 1). Recognizing these subtypes has significantly impacted medulloblastoma research and treatment, enabling more targeted and potentially effective therapies tailored to each subgroup’s unique characteristics.

The WNT subgroup, comprising approximately 10% of medulloblastoma cases, is defined by mutations in the WNT/β-catenin signaling pathway [10]. Almost all WNT subtype tumors harbor CTNNB1 mutations, leading to β-catenin activation, and are believed to originate from the lower rhombic lip of the hindbrain [14]. Notably, WNT tumors have the most favorable prognosis among the medulloblastoma subgroups, with five-year survival rates exceeding 90% [2]. This relatively good prognosis can be attributed to their reduced tendency to metastasize and high sensitivity to standard treatments, including chemotherapy and radiation [15]. In fact, WNT tumors often require less aggressive therapeutic regimens, and studies are currently exploring de-escalated treatments for WNT-medulloblastoma to minimize long-term side effects while maintaining high survival rates.

In contrast, the SHH subgroup accounts for approximately 30% of medulloblastoma cases and is characterized by the activation of the Sonic Hedgehog signaling pathway [16]. This subgroup is commonly associated with mutations in the PTCH1, SMO, and SUFU genes, which disrupt the SHH pathway and promote tumor growth [17]. SHH tumors can arise from granule neuron precursor cells in the cerebellum and occur across all age groups, including infants, children, and adults. The prognosis for SHH tumors varies depending on patient age and genetic background, with infants typically having a favorable prognosis and adults having less aggressive tumors [16]. However, SHH tumors in children between 3 and 16 years often exhibit poorer outcomes and may require intensified treatment [18]. As SHH tumors have well-defined molecular drivers, targeted therapies are being explored [19], although resistance to these treatments remains a challenge. Among the most advanced clinical candidates is vismodegib, a Smoothened (SMO) inhibitor used in SHH-subtype medulloblastoma. However, resistance frequently emerges through mutations in SMO or downstream components such as SUFU and GLI2. Clinical trials (e.g., NCT01878617) report progression in over 50% of pediatric patients within six months of treatment, highlighting the limitations of monotherapy. Additional challenges include limited drug penetration across the blood–brain barrier, off-target effects on developing tissues, and the absence of robust biomarkers to stratify responders from non-responders [20].

Group 3 medulloblastoma, representing about 25% of cases, is characterized by its association with MYC amplification and a high propensity for metastasis at diagnosis [21]. Group 3 tumors are highly aggressive and often involve genomic instability, with frequent isochromosome 17q (i17q) abnormalities and loss of chromosome 9q [22]. Consequently, Group 3 tumors have the worst prognosis among the medulloblastoma subtypes, with five-year survival rates ranging from 50% to as low as 20–30% in cases of metastasis or MYC amplification [10]. Due to their aggressiveness and resistance to conventional treatments, novel therapeutic approaches are urgently needed for this subtype. High-intensity chemotherapy and radiation are typically employed, but efforts are also being made to develop therapies targeting MYC-driven tumorigenesis, although achieving durable responses in this subtype remains challenging.

Lastly, Group 4 is the most common medulloblastoma subtype, accounting for around 35% of cases. Unlike the other subgroups, Group 4 lacks a clearly defined molecular pathway driver, but it often features recurrent chromosome 17q gain and chromosome 8 loss [23]. Although the exact cell of origin is uncertain, Group 4 tumors are believed to arise from precursor cells in the cerebellum or brainstem [13]. In terms of prognosis, Group 4 medulloblastoma has an intermediate outcome, with survival rates varying widely depending on the presence of metastasis and specific genetic features [2]. This group is heterogeneous, with some patients responding well to standard therapy, while others, particularly those with metastatic disease, face a high risk of recurrence. Research into targeted therapies for Group 4 tumors is still in its infancy, but identifying precise molecular targets could lead to more effective treatments in the future.

### 2.2. Subtype-Specific Pathways: Metabolic and Epigenetic Mechanisms

While the genetic and clinical characteristics of medulloblastoma subtypes are well documented, recent research has turned its focus to understanding the metabolic and epigenetic differences that distinguish these subtypes. These mechanisms not only shape tumor behavior but also influence prognosis and treatment response, highlighting unique vulnerabilities that could serve as therapeutic targets.

Metabolic pathways in medulloblastoma have been studied extensively, with each subgroup exhibiting distinct metabolic profiles. WNT subgroup tumors display relatively low levels of metabolic reprogramming. Preliminary studies suggest that these tumors exhibit less dependence on glycolysis and oxidative phosphorylation compared to other subtypes, contributing to their lower aggressiveness and responsiveness to conventional therapies [7]. In contrast, SHH subgroup tumors display significant metabolic reprogramming, with increased reliance on glycolysis and lipid metabolism pathways. This is supported by studies showing that SHH activation is associated with elevated glycolytic flux, oxidative phosphorylation, and fatty acid synthesis [6]. As a result, targeting these pathways could offer therapeutic opportunities, such as using fatty acid synthase inhibitors and glycolysis inhibitors, which have shown promise in preclinical studies. Furthermore, SHH pathway inhibitors like SMO antagonists have demonstrated some efficacy, but resistance often emerges through metabolic reprogramming, emphasizing the need to identify alternative metabolic targets [19]. Tumors in the Group 3 subtype, characterized by MYC amplification, exhibit pronounced metabolic alterations, including high rates of glycolysis and glutamine dependency [24]. MYC overexpression drives an aggressive metabolic phenotype, fueling rapid cell division and resistance to stress conditions. Consequently, targeting metabolic dependencies, such as glutamine metabolism and glycolysis, could disrupt tumor growth in Group 3. However, translating these findings into effective therapies remains an ongoing challenge. In contrast, metabolic studies in Group 4 medulloblastoma are relatively limited, although emerging evidence suggests that this subgroup may rely on amino acid metabolism and mitochondrial function [6]. Due to the heterogeneous nature of Group 4, metabolic vulnerabilities vary, but some patients may benefit from interventions targeting amino acid pathways or mitochondrial metabolism. Recent studies have revealed additional metabolic features of Group 4 medulloblastomas. Single-cell transcriptomic profiling has identified upregulation of branched-chain amino acid transaminases such as BCAT1, as well as enhanced expression of mitochondrial electron transfer flavoproteins ETFA and ETFB, suggesting increased oxidative metabolism in specific subpopulations. These alterations may support subtype-specific dependencies on amino acid catabolism and mitochondrial respiration. Moreover, increased expression of solute carrier transporters (SLC1A5, SLC7A5) indicates active amino acid uptake, contributing to cell growth and redox balance in Group 4 tumors [25].

In terms of epigenetic pathways, each medulloblastoma subgroup exhibits distinct profiles. WNT subgroup tumors display unique DNA methylation and histone modification profiles, which are less extensive than those found in other subtypes [10]. Epigenetic stability may contribute to the better prognosis in this group, with fewer alterations needed to maintain tumor growth. Although epigenetic modifiers have not been widely studied in this group, histone acetylation and methylation may present limited therapeutic targets for some patients [26]. Conversely, epigenetic dysregulation plays a significant role in SHH subgroup tumors, with alterations in DNA methylation and histone modifiers such as EZH2 and HDACs that promote tumor cell proliferation and maintenance of stem-like states [27]. Targeting epigenetic regulators, particularly histone methyltransferases and deacetylases, could help disrupt SHH-driven oncogenesis. For instance, inhibitors of EZH2, a histone methyltransferase, have shown promise in preclinical models by impairing the epigenetic support of SHH signaling [28]. Group 3 tumors, heavily influenced by MYC, exhibit marked epigenetic dysregulation, including widespread histone modification and DNA methylation changes [29]. MYC-driven tumors rely on chromatin remodeling enzymes and histone modifiers to maintain the high transcriptional activity associated with MYC amplification. Targeting chromatin remodeling enzymes, such as BRD4 inhibitors, has shown potential in impairing MYC-driven transcriptional programs and metabolic dependencies in Group 3 tumors [30]. Lastly, while Group 4 has fewer well-characterized epigenetic alterations, it exhibits unique DNA methylation and chromatin states associated with its genetic background [26]. Histone deacetylase (HDAC) and DNA methyltransferase inhibitors may offer therapeutic potential in this subgroup by modulating gene expression patterns [31]. However, further research is needed to clarify the role of specific epigenetic regulators in Group 4 medulloblastoma.

Each medulloblastoma subtype has a distinct profile of metabolic and epigenetic alterations, underscoring the heterogeneity and complexity of the disease. These subtype-specific differences offer potential therapeutic opportunities but also present challenges, as effective treatments must be tailored to the unique biology of each subtype. The identification of these pathways continues to expand the landscape of potential treatment targets, guiding ongoing efforts to develop precision therapies that will improve patient outcomes across all medulloblastoma subtypes. To facilitate a clearer understanding of these subtype-specific differences, we present a comparative summary of metabolic and epigenetic features across the four medulloblastoma subtypes (Table 1).

## 3. Metabolic Mechanisms in Medulloblastoma

Metabolic reprogramming is a hallmark of cancer, enabling tumor cells to sustain rapid growth, proliferation, and survival under often challenging conditions, including hypoxia and nutrient scarcity [7]. In medulloblastoma, distinct metabolic alterations have emerged as both drivers of tumor biology and potential therapeutic targets. By examining the metabolic landscape of medulloblastoma, we can gain insights into its unique vulnerabilities and how targeted interventions could improve outcomes. This section provides an in-depth exploration of the metabolic reprogramming in medulloblastoma, focusing on critical pathways and their roles across various molecular subtypes, alongside their interaction with the tumor microenvironment (Figure 2).

### 3.1. Glycolysis and the Warburg Effect

One of the most fundamental metabolic changes in medulloblastoma is the upregulation of glycolysis, which allows tumor cells to generate energy and biosynthetic precursors more rapidly than through oxidative phosphorylation alone [6]. This reliance on glycolysis, despite oxygen presence, is a hallmark of the Warburg effect, named after Otto Warburg, who first identified this phenomenon [33]. Medulloblastoma cells demonstrate high levels of glycolytic enzymes, including hexokinase 2 (HK2), pyruvate kinase M2 (PKM2), and lactate dehydrogenase A (LDHA), which support a high rate of glycolytic flux and lactic acid production [6]. Increased glycolysis provides medulloblastoma cells with ATP and metabolic intermediates essential for synthesizing nucleotides, amino acids, and lipids. The resulting accumulation of lactate acidifies the tumor microenvironment, promoting invasion and immune evasion, both of which contribute to medulloblastoma progression [34]. The upregulation of glycolysis is particularly pronounced in Group 3 medulloblastomas, known for their aggressive behavior and high MYC amplification, as MYC promotes the expression of glycolytic genes [6]. Targeting glycolysis with inhibitors of HK2, LDHA, or PKM2 has shown promise in preclinical models, indicating that disrupting glycolytic flux could be a therapeutic strategy in MYC-driven medulloblastoma [32].

### 3.2. Mitochondrial Function and Oxidative Phosphorylation

While glycolysis is heavily upregulated in medulloblastoma, mitochondrial function and oxidative phosphorylation (OXPHOS) also play crucial roles, especially in subtypes with less aggressive behavior, such as WNT and SHH medulloblastomas. Mitochondrial metabolism provides medulloblastoma cells with ATP through the electron transport chain and supplies intermediates for the tricarboxylic acid (TCA) cycle, which are essential for anabolic processes, such as nucleotide synthesis and lipid production [35]. WNT-driven tumors seem to retain more mitochondrial function and OXPHOS activity, possibly contributing to their less aggressive clinical phenotype and favorable prognosis [6]. Conversely, SHH-driven tumors display a reliance on both glycolysis and mitochondrial function, with evidence showing that SHH activation can increase OXPHOS capacity [6]. This dual metabolic phenotype in SHH tumors might support their adaptability under various environmental stresses, including changes in oxygen and nutrient availability. Therapeutic approaches aimed at inhibiting OXPHOS or mitochondrial function, such as mitochondrial uncouplers and inhibitors of the TCA cycle enzyme IDH, are being explored as potential treatments in preclinical studies.

### 3.3. Lipid Metabolism

Lipid metabolism, including both lipid synthesis and degradation pathways, is frequently altered in medulloblastoma, allowing tumor cells to generate lipid components necessary for membrane synthesis, energy storage, and signaling [6]. In highly proliferative tumors, such as Group 3 and Group 4 medulloblastomas, increased lipid synthesis is a crucial adaptation, supporting rapid cellular growth and maintaining membrane integrity [8]. The enzyme fatty acid synthase (FASN), which catalyzes de novo lipid synthesis, is often upregulated in medulloblastoma, particularly in subtypes with high proliferative indices [34]. Enhanced lipid synthesis provides not only structural components for cell membranes but also serves as an energy reservoir, supporting tumor growth under nutrient-limited conditions [36]. In addition to lipid synthesis, altered lipid degradation pathways, such as increased fatty acid oxidation (FAO), have been observed in medulloblastoma [37]. This pathway contributes to energy production and redox balance, helping tumor cells cope with oxidative stress. Targeting lipid metabolism, particularly with FASN inhibitors, is under investigation as a promising therapeutic avenue, especially for aggressive subtypes where lipid metabolism is more pronounced.

### 3.4. Amino Acid and One-Carbon Metabolism

Amino acids, such as glutamine, play a vital role in supporting medulloblastoma growth by fueling the TCA cycle, providing nitrogen for nucleotide biosynthesis, and maintaining redox balance through glutathione production [35]. Glutamine metabolism is particularly critical in MYC-amplified Group 3 tumors, where MYC enhances the expression of glutaminase (GLS), the enzyme responsible for converting glutamine to glutamate. This glutamine dependency makes Group 3 medulloblastomas especially sensitive to glutaminase inhibitors, which have shown potential in reducing tumor cell viability in preclinical studies [8]. In addition to amino acid metabolism, one-carbon metabolism is also upregulated in medulloblastoma to support DNA synthesis and methylation reactions, which are crucial for rapid cell division. One-carbon metabolism pathways involve folate and methionine cycles, which provide one-carbon units for synthesizing purines, thymidylate, and S-adenosylmethionine (SAM), a key methyl donor in epigenetic regulation [38]. Targeting one-carbon metabolism with antifolate drugs, such as methotrexate, has shown promise in medulloblastoma treatment by impairing DNA synthesis and inducing cell cycle arrest.

### 3.5. Interplay with the Tumor Microenvironment

The tumor microenvironment (TME) in medulloblastoma is shaped by and influences metabolic shifts within tumor cells. Factors such as hypoxia (low oxygen levels), nutrient availability, and interactions with stromal cells significantly impact tumor metabolism and drive further adaptations in tumor cells to survive in these challenging conditions [6]. Hypoxia, commonly found in the medulloblastoma microenvironment, upregulates hypoxia-inducible factor-1α (HIF-1α), which promotes glycolysis and suppresses OXPHOS, further driving the Warburg effect and enhancing lactate production [6]. This acidification of the TME supports tumor invasiveness, enhances immune evasion by inhibiting T-cell activity, and reinforces metabolic reprogramming. Additionally, nutrient availability in the TME can shape metabolic pathways in medulloblastoma. For instance, limited glucose or amino acids due to rapid tumor growth can activate autophagy, allowing tumor cells to recycle intracellular components to sustain metabolic needs [35]. Interactions between tumor cells and stromal cells, such as astrocytes and microglia, also contribute to medulloblastoma progression by promoting inflammation and providing metabolic support through nutrient transfer and growth factor secretion [39]. Understanding the dynamic relationship between medulloblastoma cell metabolism and the TME is essential for developing therapies that disrupt these dependencies. Strategies targeting HIF-1α, autophagy inhibitors, and approaches to normalize the TME acidity or nutrient availability are currently being investigated, with the potential to render tumor cells more vulnerable to metabolic therapies.

Medulloblastoma demonstrates a wide array of metabolic adaptations that support its growth, survival, and aggressiveness. These alterations vary across molecular subtypes, suggesting distinct metabolic vulnerabilities that could be exploited for therapy. Key metabolic pathways, such as glycolysis, mitochondrial function, lipid metabolism, and amino acid metabolism, not only fulfill the biosynthetic demands of tumor cells but also interact with the TME, shaping the overall tumor behavior and response to treatment. Targeting these metabolic pathways could potentially complement traditional therapies, offering a promising approach to improving outcomes in medulloblastoma, particularly for aggressive subtypes with limited treatment options.

## 4. Epigenetic Mechanisms in Medulloblastoma

Epigenetic mechanisms play a central role in cancer development and progression, including medulloblastoma. Unlike genetic mutations, which are fixed alterations in DNA, epigenetic changes are reversible modifications that regulate gene expression without altering the underlying DNA sequence [40]. These modifications are crucial in orchestrating complex gene expression patterns required for cellular differentiation and tumor biology. Epigenetic alterations in medulloblastoma influence tumor cell proliferation, survival, differentiation, and resistance to therapy, making them both informative biomarkers and promising therapeutic targets [40]. This section explores the primary epigenetic mechanisms at play in medulloblastoma, including DNA methylation, histone modifications, chromatin remodeling, and non-coding RNAs, with a focus on their roles in tumor initiation, maintenance, and treatment response (Figure 3).

### 4.1. DNA Methylation

DNA methylation, a crucial epigenetic mechanism, involves the addition of a methyl group to cytosine residues in CpG islands, typically resulting in gene silencing [41]. In cancer, disrupted DNA methylation patterns can lead to the inactivation of tumor suppressor genes or the activation of oncogenes [42]. Medulloblastoma exhibits varied DNA methylation profiles across its molecular subtypes, serving as both disease markers and drivers of subtype-specific biological processes [43]. Notably, different medulloblastoma subtypes display distinct DNA methylation patterns. WNT-driven tumors often maintain cell lineage and developmental pathways, contributing to their relatively favorable prognosis [44]. In contrast, SHH-subtype tumors exhibit specific DNA methylation signatures that promote SHH pathway activation, maintaining tumor cell proliferation [17]. These SHH-subtype tumors also display methylation changes in genes associated with neural development and cell cycle regulation, underscoring their differentiation status and cell-of-origin [45]. Group 3 tumors, characterized by MYC amplification and high aggressiveness, often display hypermethylation of tumor suppressor genes, supporting their highly proliferative nature and therapy resistance [46]. Group 4 tumors, although more heterogeneous, exhibit specific methylation profiles that correlate with neuronal differentiation and structural organization pathways [13]. DNA methylation alterations can be associated with poor clinical outcomes and may serve as biomarkers for prognosis. Moreover, altered DNA methylation in medulloblastoma not only facilitates tumorigenesis but also contributes to treatment resistance. By silencing genes that regulate cell death or immune response, DNA methylation patterns in aggressive subtypes of medulloblastoma can create a tumor environment that resists both chemotherapy and radiation [47]. Hence, hypomethylating agents, which can reactivate silenced tumor suppressor genes, are being explored in preclinical studies as potential therapeutic options for overcoming these methylation-driven resistances [48].

### 4.2. Histone Modifications

Histone modifications, involving the addition or removal of chemical groups (e.g., acetyl, methyl) on histone proteins around which DNA is wound, play critical roles in regulating gene expression patterns that contribute to tumor progression. These modifications affect chromatin structure and determine whether a region of DNA is accessible for transcription.

Histone acetylation and methylation are two key modifications that have been implicated in medulloblastoma. Histone acetylation, primarily at lysine residues, is generally associated with an open chromatin state and active transcription. In medulloblastoma, histone acetylation can promote the expression of oncogenes and pathways involved in tumor cell proliferation. For example, hyperacetylation of genes within the MYC pathway in Group 3 medulloblastoma correlates with their high proliferative capacity and poor prognosis [49]. In contrast, histone methylation, such as H3K27 and H3K9 methylation, can lead to either activation or repression of gene expression, depending on the location and number of methyl groups added. In SHH-subtype medulloblastomas, histone methylation is crucial for maintaining SHH signaling pathways that support tumor growth and stemness. Group 4 medulloblastomas also exhibit specific histone methylation patterns that correlate with their distinct transcriptional profiles [50].

Notably, histone modifications are dynamic and reversible, making them attractive targets for drug development. Histone deacetylase inhibitors (HDACi), which prevent the removal of acetyl groups, and histone methyltransferase inhibitors, which add methyl groups to histones, have shown promise in preclinical studies for altering the gene expression landscape in medulloblastoma, potentially limiting tumor growth and improving sensitivity to other therapies.

### 4.3. Chromatin Remodeling and Non-Coding RNAs

Chromatin remodeling refers to the structural changes in chromatin that influence gene accessibility and transcription. Chromatin remodeling complexes, such as SWI/SNF, are essential for regulating gene expression in development and differentiation. Dysregulation of these complexes in medulloblastoma can lead to abnormal activation or silencing of genes that control cell cycle, differentiation, and survival.

Medulloblastoma is characterized by aberrant chromatin regulation, which can manifest in two main ways. Firstly, chromatin remodelers such as ARID1A and SMARCA4 are often mutated or dysregulated, particularly in aggressive subtypes. These mutations can impair DNA repair processes, contributing to genomic instability and promoting tumorigenesis. Therapeutic strategies targeting chromatin remodelers are in early stages [51], but understanding their role in medulloblastoma could reveal new opportunities for targeted intervention. Secondly, non-coding RNAs (ncRNAs) play a crucial role in regulating gene expression in medulloblastoma. MicroRNAs (miRNAs) and long non-coding RNAs (lncRNAs) affect mRNA stability, translation, and chromatin organization, modulating the tumor microenvironment. For instance, specific miRNAs, such as miR-17/92, are overexpressed in medulloblastoma and promote cell cycle progression by suppressing tumor suppressor genes. Additionally, lncRNAs, such as HOTAIR, are implicated in medulloblastoma progression by modulating chromatin states and facilitating transcriptional activation of oncogenic pathways.

Non-coding RNAs serve as potential biomarkers due to their stability in biological fluids [52], making them promising candidates for non-invasive diagnostics in medulloblastoma. Therapeutic targeting of miRNAs and lncRNAs is being explored, aiming to restore normal gene expression patterns and reduce tumor cell proliferation.

### 4.4. Epigenetic Heterogeneity and Tumor Adaptation

Epigenetic heterogeneity, the ability of cells to reversibly alter their gene expression profiles in response to environmental changes, plays a crucial role in medulloblastoma’s capacity to adapt, survive, and resist therapy. Under the selective pressure of treatments like chemotherapy and radiotherapy, tumor cells can dynamically alter their epigenetic landscape [53]. For instance, aggressive subtypes of medulloblastoma may silence pro-apoptotic genes via DNA methylation or activate repair pathways through histone modifications, contributing to therapy resistance and tumor recurrence [54]. The adaptability of medulloblastoma cells to evade treatment is a significant challenge in effective therapy. Epigenetic modifications enable these cells to silence DNA damage response genes or activate pathways that promote survival, leading to residual tumor cells that survive initial treatments and eventually cause relapse. This is an area where targeting epigenetic regulators could potentially resensitize these resistant cells to conventional therapies [55]. Moreover, epigenetic mechanisms also allow medulloblastoma cells to interact with and shape the tumor microenvironment. For example, hypoxia can induce epigenetic changes that promote angiogenesis and immune evasion, further supporting tumor growth and metastasis [56]. These interactions underscore the importance of epigenetics in medulloblastoma not only as an intrinsic regulator of tumor cell behavior but also as a means for cells to adapt to and modify their surroundings. In inhibiting DNA methylation, HDACs, and chromatin remodelers, epigenetic therapies represent a promising approach to counteracting medulloblastoma’s adaptability and overcoming therapy resistance [57]. By targeting the reversible nature of epigenetic changes, these therapies could limit tumor plasticity, potentially improving long-term treatment efficacy and patient outcomes. Overall, epigenetic mechanisms profoundly influence medulloblastoma biology by regulating key processes that drive tumor initiation, progression, and therapy response. These epigenetic alterations offer potential therapeutic opportunities that could complement or enhance current treatments, from DNA methylation and histone modifications to chromatin remodeling and non-coding RNAs. Understanding the specific epigenetic landscape of medulloblastoma subtypes and their roles in tumor plasticity provides a foundation for developing targeted interventions, which could improve prognosis and reduce recurrence rates in patients with medulloblastoma.

## 5. Interconnection Between Metabolic and Epigenetic Pathways

The interplay between metabolic and epigenetic pathways is increasingly recognized as a driving force behind cancer development and progression, including medulloblastoma. Metabolic and epigenetic mechanisms in medulloblastoma do not operate independently; instead, they form a complex network of feedback loops and interdependencies. Metabolites generated from altered metabolic pathways can serve as substrates or cofactors for epigenetic enzymes, directly impacting gene expression. Conversely, epigenetic modifications can regulate the expression of metabolic enzymes, thereby modulating cellular metabolism to meet the demands of a rapidly proliferating tumor. This interconnection between metabolic and epigenetic pathways is central to understanding how medulloblastoma cells adapt to their environment, evade therapeutic interventions, and sustain tumor growth (Figure 4).

### 5.1. Metabolic Regulation of Epigenetics

The metabolic state of a cell influences epigenetic modifications, as many metabolites act as direct substrates or cofactors for epigenetic enzymes. For instance, altered levels of acetyl-CoA, α-ketoglutarate, S-adenosylmethionine (SAM), and NAD+ can impact histone acetylation, DNA methylation, and chromatin remodeling [58]. In medulloblastoma, these metabolites are often dysregulated, contributing to an epigenetic environment that promotes oncogenesis and tumor maintenance. One key player in this process is acetyl-CoA, a critical metabolite in cancer metabolism often elevated due to increased glycolytic flux and enhanced lipid synthesis [59]. Acetyl-CoA serves as the substrate for histone acetyltransferases (HATs), which add acetyl groups to histone tails, resulting in chromatin relaxation and active transcription. In medulloblastoma, elevated levels of acetyl-CoA promote the acetylation of histones associated with oncogenes, particularly in aggressive subtypes such as Group 3 [60]. The availability of acetyl-CoA is therefore directly linked to gene expression patterns that support cell proliferation, resistance to apoptosis, and stemness. Another important metabolite is α-ketoglutarate, a cofactor for the TET family of DNA demethylases and Jumonji-C domain-containing histone demethylases (KDMs), both of which are involved in the removal of methyl groups from DNA and histones, respectively [61]. This demethylation allows for the reactivation of genes involved in cellular differentiation, which is often suppressed in cancer. In medulloblastoma, dysregulation of α-ketoglutarate levels can influence TET activity, affecting DNA methylation patterns and promoting a stem-cell-like state that supports tumor growth and therapy resistance. Furthermore, S-adenosylmethionine (SAM) is the primary methyl donor in cells, essential for DNA and histone methylation [62]. Tumors often exhibit altered SAM metabolism, leading to hypermethylation of tumor suppressor genes and the silencing of differentiation pathways. This alteration is particularly relevant in WNT and SHH subtypes of medulloblastoma, where hypermethylation of tumor suppressor genes correlates with poor prognosis and aggressive behavior [63]. By influencing SAM availability, metabolic pathways can shape the DNA methylation landscape, facilitating a tumor-permissive environment. Additionally, NAD+ is required for the activity of sirtuins, a family of histone deacetylases that play roles in tumor suppression and cellular stress responses [64]. Elevated NAD+ levels can enhance the deacetylation of tumor suppressor genes, which may facilitate cellular resilience under metabolic stress. In medulloblastoma, NAD+ availability is often dysregulated, affecting sirtuin activity and contributing to a regulatory environment that favors tumor growth.

Through these metabolic intermediates, medulloblastoma cells are able to fine-tune epigenetic modifications to meet the demands of tumor progression. Understanding these connections offers therapeutic opportunities, as altering metabolite levels may provide a way to influence epigenetic states and potentially disrupt tumor-promoting gene expression patterns.

### 5.2. Epigenetic Regulation of Metabolism

Epigenetic mechanisms also play a crucial role in regulating the expression of genes involved in key metabolic pathways in medulloblastoma. This regulation is essential for tumor cells to adapt their metabolism to their specific growth requirements. For instance, controlling the accessibility of genes that encode metabolic enzymes enables medulloblastoma cells to upregulate pathways that promote biomass production and energy generation, while downregulating pathways that would hinder rapid growth. DNA methylation, histone modification, and non-coding RNAs are key epigenetic mechanisms that modulate metabolic pathways in medulloblastoma. DNA methylation, for example, can silence or activate genes involved in metabolic processes. Hypermethylation of genes involved in oxidative phosphorylation has been observed in highly glycolytic tumors [65], supporting the Warburg effect and allowing medulloblastoma cells to sustain high proliferation rates. Conversely, hypomethylation of genes involved in lipid synthesis promotes lipid accumulation, which is necessary for membrane synthesis in rapidly dividing cells [66]. Histone modifications, particularly acetylation and methylation, also regulate metabolic genes by altering chromatin accessibility. In medulloblastoma, histone acetylation at promoter regions of glycolytic enzymes, such as hexokinase and pyruvate kinase, enhances glycolysis in tumors [35]. Additionally, histone methylation of genes involved in mitochondrial function can limit oxidative metabolism, promoting a shift towards glycolysis and lactate production. These adaptations are advantageous in the hypoxic tumor microenvironment. Furthermore, non-coding RNAs, including miRNAs and lncRNAs, modulate metabolic pathways by targeting mRNAs of metabolic enzymes. The upregulation of miR-17/92 in medulloblastoma, for instance, targets multiple metabolic genes, supporting the metabolic switch towards glycolysis [67]. Long non-coding RNAs, such as HOTAIR, also impact metabolism by recruiting chromatin remodelers to silence or activate metabolic genes, shaping the metabolic phenotype of the tumor [68]. By modulating the expression of metabolic enzymes, these epigenetic changes create an environment that promotes tumor survival and proliferation. These adaptations allow tumors to optimize their metabolism for growth and adapt to the specific nutrient and oxygen constraints of the tumor microenvironment.

### 5.3. Metabolic and Epigenetic Crosstalk in MB

The interconnection between metabolic and epigenetic pathways is not a one-way street; instead, they are linked through feedback loops and crosstalk that reinforce each other, creating a stable environment conducive to tumor growth. A positive feedback loop involving glycolysis and histone acetylation is an example of this interconnection [69]. In medulloblastoma, upregulation of glycolysis leads to increased acetyl-CoA production, which in turn promotes histone acetylation and activates genes involved in glycolysis, creating a self-sustaining loop [35]. This loop not only maintains high glycolytic activity but also facilitates the epigenetic environment needed for continued proliferation. Moreover, the availability of S-adenosylmethionine (SAM), influenced by methionine metabolism, can enhance DNA and histone methylation patterns that silence tumor suppressor genes, further promoting tumorigenesis [62]. Another key aspect of metabolic-epigenetic crosstalk is the regulation of metabolite-sensitive enzymes, such as TETs and KDMs, by α-ketoglutarate and other metabolic intermediates [69]. In medulloblastoma, metabolic shifts that decrease α-ketoglutarate levels can inhibit these enzymes, leading to hypermethylation and gene silencing. This suppression can stabilize a gene expression program that supports undifferentiated, stem-cell-like properties in tumor cells, creating an environment favorable for tumor persistence and progression [70]. For instance, hypoxia, a common feature of solid tumors, can alter cellular metabolism and reduce the availability of key metabolites (such as oxygen-dependent α-ketoglutarate), inhibiting the function of TET enzymes and promoting DNA hypermethylation [71]. Furthermore, hypoxic conditions also induce HIF-1α, a transcription factor that activates glycolytic genes, further supporting a feedback loop that maintains high glycolysis and reduces oxidative phosphorylation [72]. This metabolic-epigenetic feedback loop enables medulloblastoma cells to thrive even in low-oxygen environments, enhancing survival and resistance to therapy. Additionally, the interdependence between metabolic and epigenetic pathways contributes to the adaptive capacity of medulloblastoma, especially under therapeutic pressure. For example, treatments that target glycolysis can lead to metabolic reprogramming that shifts tumor cells toward oxidative phosphorylation, accompanied by epigenetic changes that activate mitochondrial genes [35]. Similarly, epigenetic therapies that reduce DNA methylation or alter histone acetylation can induce metabolic reprogramming, enabling tumor cells to find alternative nutrient sources and evade treatment [73]. This adaptability highlights the challenge of targeting metabolic and epigenetic pathways in isolation and underscores the potential benefit of combination therapies. The interdependence between metabolic and epigenetic pathways in medulloblastoma establishes a robust regulatory network that supports tumor growth, adaptation, and resistance to therapy. Direct evidence for such feedback has been demonstrated in MYC-driven medulloblastoma models, where increased glycolytic flux elevates intracellular acetyl-CoA levels, enhancing histone H3K27 acetylation and upregulating proliferation-associated genes [74]. Additionally, glutamine-derived α-ketoglutarate promotes TET-mediated DNA demethylation, sustaining expression of neural stem cell markers. These examples support a bidirectional relationship, where metabolic intermediates both respond to and shape epigenetic states, reinforcing oncogenic programs [74]. Understanding the specifics of these feedback loops and crosstalk mechanisms not only provides deeper insight into medulloblastoma biology but also reveals opportunities for targeting these interconnected pathways in a combined therapeutic approach. By disrupting both the metabolic drivers and the epigenetic regulators of tumor cell proliferation, future treatments may be able to more effectively halt tumor progression and improve patient outcomes.

## 6. Current Therapeutic Approaches Targeting Metabolic and Epigenetic Mechanisms

Advances in understanding the metabolic and epigenetic underpinnings of medulloblastoma have opened new avenues for therapeutic intervention. Targeted therapies designed to interfere with key metabolic and epigenetic pathways aim to disrupt the processes critical for tumor growth and survival. As medulloblastoma tumors often exhibit complex metabolic and epigenetic dependencies that vary across subtypes, a variety of targeted strategies have been explored. Here, we examine the latest in metabolic and epigenetic therapies and discuss the potential and challenges associated with combination treatments that target both of these pathways simultaneously (Figure 5).

### 6.1. Metabolic Therapies

Altered metabolic processes, such as enhanced glycolysis, increased fatty acid synthesis, and dysregulated amino acid metabolism, are central to medulloblastoma pathology. This metabolic reprogramming supports the high energy and biosynthetic demands of rapidly proliferating tumor cells.

Glycolysis inhibitors, for example, have shown potential in preclinical studies. Medulloblastoma cells, like many tumors, often display the Warburg effect, where they rely heavily on glycolysis even in the presence of oxygen. 2-Deoxy-D-glucose (2-DG) is a glycolysis inhibitor that competitively inhibits glucose metabolism. Although preclinical studies have shown promising effects in reducing tumor growth, clinical trials are needed to further evaluate its efficacy and safety in pediatric medulloblastoma patients [75]. Another promising compound is lonidamine, an inhibitor of the glycolytic enzyme hexokinase, which has demonstrated potential in preclinical models of brain tumors by effectively impairing ATP production and inducing apoptosis in cancer cells [76]. In addition to glycolysis inhibitors, fatty acid synthase (FASN) inhibitors also hold promise. The upregulation of lipid biosynthesis is a hallmark of many cancers, including medulloblastoma, where fatty acids are crucial for membrane synthesis and signaling. Orlistat, an FASN inhibitor initially approved as an anti-obesity drug, has shown anti-tumor activity in preclinical medulloblastoma models by blocking fatty acid synthesis and inducing cell death [35]. Other FASN inhibitors, such as TVB-2640, are also under clinical investigation, with early data indicating potential benefits in solid tumors [77]. Since lipid metabolism is less active in normal brain cells compared to medulloblastoma cells, FASN inhibitors hold promise as selective metabolic interventions for targeting tumor cells [37]. Moreover, glutaminase inhibitors and mitochondrial function modulators are also being explored. Medulloblastoma cells require a steady supply of glutamine for energy and biosynthesis. CB-839, a glutaminase inhibitor, targets glutamine metabolism and has shown efficacy in preclinical studies across various cancers by depriving cells of a critical substrate needed for the TCA cycle and biosynthetic reactions [78]. Trials in solid tumors, including brain cancers, are ongoing, and if successful, CB-839 could offer a viable metabolic target in medulloblastoma [60]. This approach may be particularly relevant in aggressive subtypes that rely on glutamine metabolism for rapid growth. Furthermore, mitochondrial oxidative phosphorylation (OXPHOS) is another potential target, particularly for medulloblastoma subtypes with active mitochondrial metabolism. Drugs like metformin and phenformin, which disrupt mitochondrial complex I, have shown potential in preclinical studies to reduce ATP production and hinder tumor growth [34]. Although primarily explored in other cancers, mitochondrial function modulators represent a promising line of investigation for medulloblastoma, especially in combination with agents targeting glycolysis.

While these metabolic inhibitors have demonstrated potential in preclinical settings, challenges remain in translating them to clinical use due to concerns about toxicity, especially in pediatric patients. Nevertheless, advancements in precision medicine may allow for more tailored applications of these therapies, minimizing risks while maximizing therapeutic impact.

### 6.2. Epigenetic Therapies

Epigenetic modifications play a critical role in medulloblastoma progression by regulating gene expression profiles that support tumor growth and stemness. As a result, drugs targeting these modifications aim to reverse the cancer-promoting epigenetic landscape, and several have reached clinical trials. One promising therapeutic approach is the use of DNA methylation inhibitors, which target DNA methyltransferase (DNMT), an enzyme responsible for silencing tumor suppressor genes and maintaining cancerous states [79]. Preclinical studies have shown that DNMT inhibitors, such as 5-azacytidine and decitabine, can potentially reactivate silenced tumor suppressor genes, impairing tumor growth. Although limited data are available in medulloblastoma specifically, clinical trials focusing on pediatric brain tumors may provide more insight into the utility of DNMT inhibitors in this context.

Histone deacetylase (HDAC) inhibitors also present a valuable therapeutic option, as they can modify chromatin structure and reactivate tumor suppressor genes. Vorinostat (SAHA) and panobinostat, for example, have demonstrated anti-tumor effects in preclinical studies of medulloblastoma, mainly through induction of apoptosis and reactivation of tumor suppressor genes [80]. Panobinostat, in particular, has shown promise in preclinical pediatric brain tumor models, and early-stage trials are exploring its efficacy in combination with other therapies in medulloblastoma patients. Another emerging therapeutic strategy involves targeting BET bromodomain proteins, which regulate transcription of oncogenes by recognizing acetylated histones. BET inhibitors, such as JQ1, can effectively disrupt the transcription of genes crucial for cell growth and survival, and preclinical studies have shown that they can reduce tumor cell proliferation in medulloblastoma and sensitize cells to other therapeutic agents [81]. Additionally, non-coding RNAs and microRNAs (miRNAs) are increasingly recognized as epigenetic regulators in medulloblastoma. miR-17-92, for instance, is known to promote tumor growth by regulating multiple oncogenic pathways [82]. Therapeutic strategies targeting miRNA expression, such as anti-miR oligonucleotides, are under investigation and could provide a novel approach to disrupting oncogenic signals in medulloblastoma.

In conclusion, epigenetic therapies hold substantial promise for medulloblastoma, but challenges persist in terms of specificity and minimizing off-target effects, especially in developing brains. However, given the adaptability of epigenetic changes, targeting these pathways may be particularly beneficial in addressing treatment resistance.

### 6.3. Combination Therapies

Given the interplay between metabolic and epigenetic pathways, combination therapies that target both may produce synergistic effects in medulloblastoma treatment. This dual targeting approach leverages the dependency of metabolic processes on epigenetic modifications, and vice versa, creating a multi-pronged attack on tumor survival mechanisms.

Combination therapies aim to exploit vulnerabilities created by the crosstalk between metabolism and epigenetics. For example, glycolysis inhibitors combined with HDAC inhibitors can impair ATP production and simultaneously modify chromatin structure to promote tumor cell death [83]. Preclinical studies have shown that combining DNMT or HDAC inhibitors with metabolic inhibitors can have enhanced anti-tumor effects. For instance, combining CB-839 (a glutaminase inhibitor) with decitabine has demonstrated improved efficacy in certain cancers, suggesting that similar strategies may be effective in medulloblastoma. Likewise, combining glycolysis inhibitors with HDAC inhibitors has been shown to increase tumor cell sensitivity to apoptosis in models of brain tumors [84]. However, implementing these approaches in a clinical setting presents challenges, particularly regarding toxicity and patient tolerance. Metabolic and epigenetic pathways are also essential for normal cellular function, so targeting both can lead to significant side effects. Careful dosing, patient monitoring, and refinement of delivery methods will be crucial to maximizing efficacy while minimizing harm.

Combination therapies may offer a path forward in medulloblastoma treatment, especially for subtypes that are resistant to standard therapies. By disrupting both metabolic and epigenetic dependencies, it may be possible to achieve more durable responses and reduce the likelihood of relapse. The therapeutic landscape for medulloblastoma is expanding with new metabolic and epigenetic targets. Although individual therapies targeting these pathways have shown promise, combination strategies that address the interconnected nature of metabolism and epigenetics may yield even greater efficacy. In preclinical models of Group 3 medulloblastoma, combination therapy with 2-deoxyglucose (2-DG) and the histone deacetylase inhibitor vorinostat led to a >60% reduction in tumor volume compared to 20–30% with monotherapy. This combination also significantly increased apoptosis, as evidenced by cleaved caspase-3 staining and TUNEL assays [85]. These findings suggest synergistic potential by simultaneously disrupting energy metabolism and chromatin structure [85]. Continued research into the mechanisms linking metabolic and epigenetic pathways will be essential in designing targeted and effective treatment options, with the potential to improve outcomes and reduce treatment resistance in medulloblastoma patients.

## 7. Challenges and Future Directions

As research into medulloblastoma’s metabolic and epigenetic mechanisms advances, new therapeutic opportunities have emerged, yet several challenges remain. Given the aggressive nature of medulloblastoma and its distinct subtypes, translating scientific discoveries into effective treatments is complex. Here, we examine critical challenges facing metabolic and epigenetic therapies and highlight promising areas for future research that could lead to more effective, personalized treatments.

### 7.1. Heterogeneity in Tumor Response

Medulloblastoma is highly heterogeneous, with distinct molecular subtypes (WNT, SHH, Group 3, and Group 4), each exhibiting unique biological characteristics, genetic profiles, and responses to treatment. This diversity influences how tumors respond to metabolic and epigenetic therapies, as specific alterations may vary significantly across subtypes and even between patients within the same subtype.

Tumor heterogeneity arises from two main sources: intrinsic and extrinsic factors. Genetic, epigenetic, and microenvironmental differences within the tumor itself, as well as interactions with the surrounding microenvironment, influence therapeutic efficacy. These differences are evident in the distinct molecular subtypes. For example, SHH-subtype medulloblastomas are often driven by mutations affecting the SHH pathway, leading to differential sensitivity to therapies targeting oxidative phosphorylation or lipid metabolism. Similarly, WNT-subtype tumors may be more responsive to therapies that leverage metabolic vulnerabilities specific to WNT signaling [86]. Understanding these nuances is crucial for designing effective subtype-specific treatment approaches.

Moreover, therapeutic resistance and adaptive mechanisms pose a significant challenge in medulloblastoma treatment. Metabolic and epigenetic adaptations allow tumors to develop resistance to targeted therapies over time. For instance, metabolic plasticity enables medulloblastoma cells to switch between energy sources, reducing the efficacy of single-target metabolic inhibitors [6]. Additionally, epigenetic heterogeneity allows tumors to reprogram gene expression in response to therapeutic pressure, thereby evading initial responses. Overcoming this adaptive resistance requires a deeper understanding of how tumors exploit metabolic and epigenetic flexibility, particularly in response to therapeutic interventions.

### 7.2. Biomarker Development

To enhance therapeutic efficacy and guide personalized treatment, the identification and validation of robust biomarkers are essential. Biomarkers can help predict how patients will respond to specific therapies, allowing for more tailored approaches that minimize ineffective treatments and unnecessary side effects. The development of predictive biomarkers for metabolic and epigenetic therapies holds great promise for improving patient outcomes in medulloblastoma. Identifying biomarkers that reflect specific metabolic or epigenetic dependencies in medulloblastoma subtypes could inform patient stratification and therapy selection. For example, levels of metabolic intermediates such as lactate or acetyl-CoA, or epigenetic markers like DNA methylation status, may help identify patients who are likely to benefit from glycolysis or DNA methyltransferase inhibitors, respectively. Additionally, biomarkers that indicate tumor reliance on specific amino acids (e.g., glutamine) or lipid pathways could guide metabolic inhibitor use, increasing treatment precision [35]. In addition to predicting treatment response, dynamic biomarkers can also capture changes in response to treatment, enabling real-time monitoring of therapeutic efficacy. Epigenetic markers, such as histone modification patterns or microRNA signatures, may serve as indicators of treatment response, while metabolic imaging biomarkers could reveal shifts in energy use or nutrient uptake. To develop such biomarkers, longitudinal studies are needed to understand how these factors change over the course of treatment, ultimately enabling better prediction of treatment outcomes. Despite the promise of biomarkers, challenges persist in validating them for clinical use. Biomarker variability between patients and subtypes, as well as the need for minimally invasive collection methods (e.g., liquid biopsies), complicates their development and standardization. However, addressing these challenges could significantly improve the specificity and sensitivity of medulloblastoma therapies, leading to improved patient outcomes [87].

### 7.3. Potential for Immunometabolism and Immunoepigenetics

Emerging fields such as immunometabolism and immunoepigenetics offer new directions for medulloblastoma therapy, focusing on how metabolic and epigenetic pathways influence immune responses within the tumor microenvironment.

Metabolic modulation of immune cells is a promising area of research, as tumor cells often compete with immune cells for metabolic resources, such as glucose and amino acids, creating an immunosuppressive environment that hinders anti-tumor immunity. By reprogramming these pathways, researchers could restore immune function and enhance immune-mediated tumor clearance. For instance, targeting metabolic checkpoints like lactate production could prevent immunosuppressive conditions, making the tumor microenvironment more conducive to immune cell activity [88]. Furthermore, epigenetic mechanisms, such as DNA methylation and histone modifications, can modulate the expression of genes involved in immune recognition and response. This regulation affects not only tumor cells but also immune cells within the tumor microenvironment, such as T cells and macrophages. Epigenetic therapies like HDAC inhibitors have been shown to enhance immune response by upregulating immune-related genes, potentially increasing tumor immunogenicity and improving the efficacy of immunotherapies [89].

In medulloblastoma, which often evades immune surveillance, integrating metabolic and epigenetic approaches with immunotherapies may create synergistic opportunities. For example, inhibiting metabolic pathways that support immunosuppressive cells (e.g., regulatory T cells) could enhance the efficacy of checkpoint inhibitors or CAR-T therapies in medulloblastoma. In addition, medulloblastoma is widely recognized as an “immune-cold” tumor, characterized by low immune cell infiltration, low tumor mutational burden, and minimal neoantigen presentation, all of which contribute to resistance against immune checkpoint inhibitors [26,90]. However, recent studies suggest that epigenetic reprogramming could overcome this resistance by enhancing tumor immunogenicity. For example, DNA demethylating agents and HDAC inhibitors have been shown to upregulate MHC class I expression, antigen processing machinery, and interferon-stimulated genes in pediatric brain tumor models, including glioma and medulloblastoma [26,90]. These agents may increase tumor visibility to cytotoxic T lymphocytes and natural killer cells. Similarly, metabolic interventions that reduce lactate accumulation or modulate tryptophan/kynurenine metabolism may alleviate local immunosuppression, restoring T cell activation and improving CAR-T cell persistence [91]. These findings highlight a promising role for combining epigenetic or metabolic therapies with immunotherapy, especially in MYC-amplified Group 3 tumors where immune evasion mechanisms are pronounced [92]. Future work should investigate how reprogramming the tumor’s epigenetic and metabolic landscape can prime the medulloblastoma microenvironment for successful checkpoint blockade or CAR-T engagement. However, integrating these approaches requires a better understanding of how metabolic and epigenetic regulation affects immune dynamics within pediatric brain tumors, an area that remains underexplored [93].

### 7.4. Preclinical Models and Translational Studies

Robust preclinical models and translational research are essential for validating therapeutic targets and ensuring that findings can be effectively translated to clinical settings. However, gaps in preclinical research limit the ability to assess the efficacy and safety of metabolic and epigenetic therapies in medulloblastoma.

The limitations of current models, such as traditional cell line models and xenografts, may not fully capture the complexity of medulloblastoma’s metabolic and epigenetic landscape, especially in the context of pediatric brain tumors. In contrast, patient-derived xenografts (PDX) and organoid models are promising tools that better mimic the heterogeneity of primary tumors, allowing researchers to study the metabolic and epigenetic dependencies of distinct medulloblastoma subtypes. Recent single-cell RNA sequencing has revealed that Group 4 medulloblastomas comprise at least three subpopulations with distinct transcriptional signatures resembling glutamatergic neurons, progenitors, or uncommitted precursors [25]. Complementary cerebellar organoid models engineered with KDM6A or ZNF292 mutations reproduce epigenetic dysregulation seen in patient tumors, supporting their utility in studying developmental and metabolic heterogeneity. These advances provide crucial tools for resolving Group 4 complexity and identifying new therapeutic targets. These models can facilitate testing of targeted therapies and help elucidate the mechanisms of resistance observed in clinical settings.

Recent innovations in 3D culture systems and brain organoid models offer innovative platforms for studying tumor metabolism, epigenetics, and drug responses [94]. By recapitulating the tumor microenvironment more accurately than traditional cultures, these models enable researchers to examine how metabolic and epigenetic changes interact with the microenvironment, revealing insights into tumor behavior and therapeutic response.

To bridge the gap between laboratory findings and clinical applications, translational studies that validate potential therapies in relevant models are necessary. Collaborative efforts, such as multi-center studies and integration of pediatric brain tumor consortia, could accelerate the translation of metabolic and epigenetic targets from bench to bedside. These studies should aim to assess not only efficacy but also safety and tolerability, especially given the sensitivity of pediatric patients to off-target effects.

Furthermore, conducting clinical trials in pediatric populations is particularly challenging due to ethical considerations, limited patient numbers, and the need for treatments that minimize long-term side effects [95]. Innovative trial designs, such as adaptive trials and basket trials, could enable more efficient assessment of targeted therapies across diverse medulloblastoma subtypes. Integrating pharmacogenomic and biomarker data into these trials may also facilitate precision medicine approaches that better address the heterogeneous nature of medulloblastoma.

In conclusion, future research in medulloblastoma should prioritize addressing the challenges of tumor heterogeneity, refining biomarker development, and exploring the interplay between metabolic and epigenetic mechanisms within the tumor and its immune microenvironment. Collaborative initiatives between basic scientists, clinicians, and translational researchers will be essential to advance these efforts and develop treatments that improve survival while minimizing long-term impacts for pediatric patients.

## 8. Discussion

This review underscores the central role of metabolic and epigenetic mechanisms in medulloblastoma (MB), an aggressive and heterogeneous pediatric brain tumor. We have examined how the four major molecular subtypes—WNT, SHH, Group 3, and Group 4—exhibit distinct metabolic dependencies and epigenetic regulatory patterns that govern tumor progression, therapeutic response, and prognosis. A comprehensive understanding of these subtype-specific mechanisms provides a platform for developing targeted therapies and optimizing patient outcomes.

Medulloblastoma subtypes rely on unique metabolic programs: WNT tumors show elevated glycolysis, consistent with the Warburg effect, while SHH tumors exhibit reliance on oxidative phosphorylation and mitochondrial metabolism [6,66]. Group 3 MB, particularly MYC-amplified subtypes, upregulate de novo serine and glycine synthesis pathways, and Group 4 tumors exploit amino acid and nucleotide biosynthetic processes [13,21]. Dysregulated lipid metabolism, including fatty acid synthesis and cholesterol biosynthesis, supports proliferation and immune evasion across several subtypes [36,37].

Epigenetically, MB exhibits distinct DNA methylation profiles, histone modification signatures, and chromatin remodeling activities that mirror developmental origins and therapeutic vulnerabilities [26,43,47]. Subtype-specific DNA methylation landscapes have proven useful for classification and risk stratification in clinical practice [47], and histone deacetylase (HDAC) or methyltransferase inhibitors have shown potential for reversing oncogenic transcriptional programs [28,29].

The interplay between metabolism and epigenetics forms a dynamic regulatory network where metabolic intermediates—such as acetyl-CoA, α-ketoglutarate, and S-adenosylmethionine—modulate the activity of chromatin regulators, DNA/histone modifiers, and transcriptional programs [58,69]. For instance, the availability of α-ketoglutarate influences the function of TET enzymes involved in DNA demethylation, directly linking mitochondrial metabolism to epigenetic reprogramming [61].

Therapeutically, targeting these metabolic and epigenetic circuits holds promise. Glycolysis inhibitors like 2-deoxyglucose, inhibitors of fatty acid synthase (FASN), and glutaminase inhibitors are being explored in preclinical MB models [6,21,83]. Epigenetic drugs, such as DNMT inhibitors (e.g., decitabine) and HDAC inhibitors (e.g., vorinostat), have been tested in glioblastoma and pediatric brain tumor trials and are under investigation for MB [28,79,80]. Combination therapies addressing both metabolic and epigenetic pathways may prevent resistance and improve durability of response, especially in high-risk subtypes such as Group 3 [60,92].

Importantly, a clear roadmap for preclinical-to-clinical translation is needed. First, patient-derived xenografts (PDX) and patient-derived organoids (PDOs) have emerged as robust models to recapitulate MB subtype-specific features and test therapeutic responses in a physiologically relevant context [94]. These models preserve the heterogeneity of the original tumors and provide a platform for screening epigenetic and metabolic agents, both alone and in combination with immunotherapies. Integration of single-cell transcriptomic data can further refine these models by revealing subtype-specific vulnerabilities and treatment-adaptive pathways [25].

Second, biomarker-guided stratification is crucial for translating these findings into precision medicine. DNA methylation signatures, MYC/MYCN amplification, metabolic enzyme expression (e.g., PHGDH, GLS), and epigenetic modulator profiles (e.g., EZH2, BRD4) can serve as predictive biomarkers for therapeutic sensitivity [10,21,47]. Liquid biopsies based on circulating tumor DNA or non-coding RNAs may allow real-time monitoring of treatment efficacy and resistance [52,87]. Future trials should incorporate such biomarkers for patient enrollment and treatment response evaluation.

Several clinical trials already incorporate elements of this translational approach. The St. Jude MB clinical protocol (NCT01878617) stratifies patients based on molecular features and integrates novel agents targeting pathways like SHH and MYC [20]. Other trials are evaluating HDAC inhibitors in combination with radiation or chemotherapy (e.g., NCT04315064, NCT04351380), while studies in gliomas or pediatric high-grade gliomas may offer insights transferable to MB [18,28].

Emerging areas such as immunometabolism and immunoepigenetics also warrant deeper exploration. MB’s immune-cold microenvironment limits the success of checkpoint blockade therapies [90]. However, epigenetic and metabolic modulation may sensitize tumors to immune infiltration. HDAC inhibitors and DNMT inhibitors can upregulate MHC class I molecules and interferon-stimulated genes, enhancing antigen presentation and T-cell engagement [11,26]. Simultaneously, metabolic reprogramming to reduce tumor lactate or rewire tryptophan metabolism may counteract immunosuppression and improve CAR-T cell function [67,88,93]. These strategies are particularly relevant for Group 3 MB, which displays high MYC activity and marked immune evasion [5,13].

To advance the field, future research should prioritize three goals: (1) development of high-fidelity preclinical models that reflect the molecular complexity of each MB subtype; (2) rigorous identification and validation of metabolic and epigenetic biomarkers to guide therapy; and (3) design of combination strategies that integrate targeted agents with immunotherapies. Cross-disciplinary collaborations involving genomics, metabolism, epigenetics, and immunology will be essential to translate molecular findings into durable clinical responses.

In conclusion, targeting metabolic and epigenetic mechanisms offers a multifaceted approach to treating medulloblastoma. The path forward lies in integrating subtype-specific biology, leveraging patient-derived models, and implementing biomarker-driven clinical trial designs. By embracing this translational roadmap, we can refine therapies, overcome resistance, and ultimately improve survival and quality of life for children with this challenging disease.

## Figures and Tables

**Figure 1 biomedicines-13-01898-f001:**
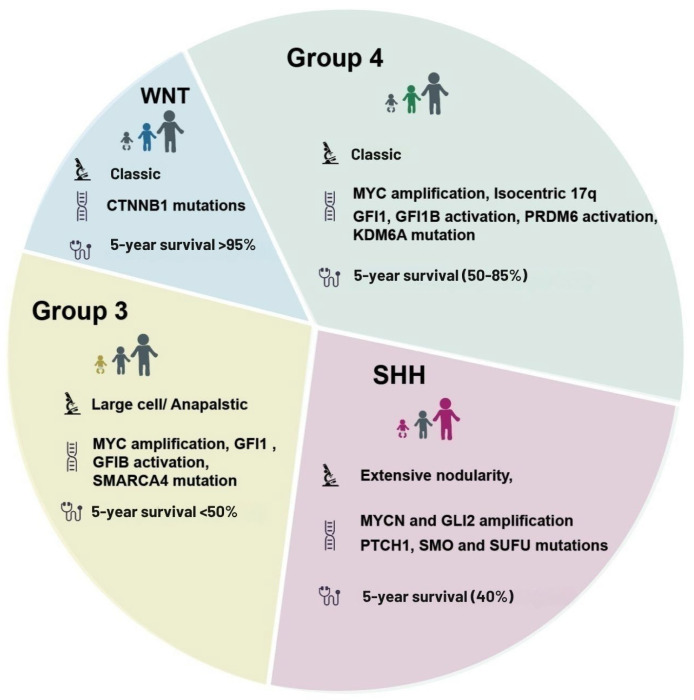
Molecular subgroups of medulloblastoma: genetic alterations, histological features, and prognosis. Schematic representation of the four major molecular subgroups of medulloblastoma: WNT, SHH (Sonic Hedgehog), Group 3, and Group 4. Each segment illustrates representative histological variants, predominant genetic alterations, and estimated 5-year overall survival rates. The WNT subgroup is characterized by classic histology, CTNNB1 mutations, and favorable prognosis (>95% 5-year survival). The SHH subgroup presents with extensive nodularity and includes MYCN and GLI2 amplification, as well as mutations in PTCH1, SMO, and SUFU, with an intermediate prognosis (~40% 5-year survival). Group 3 typically displays large cell/anaplastic histology, MYC amplification, and mutations in GFI1, GF1B, and SMARCA4, correlating with poor prognosis (<50% 5-year survival). Group 4, the most heterogeneous, is associated with classic histology, MYC amplification, isochromosome 17q, activation of GFI1 and PRDM6, and KDM6A mutations, with variable outcomes (50–85% 5-year survival). Abbreviations: CTNNB1, catenin beta 1; MYCN, MYCN proto-oncogene; GLI2, GLI family zinc finger 2; PTCH1, patched 1; SMO, smoothened; SUFU, suppressor of fused homolog; SMARCA4, SWI/SNF related, matrix associated, actin-dependent regulator of chromatin subfamily A member 4; KDM6A, lysine demethylase 6A; GFI1/B, growth factor independent 1/B.

**Figure 2 biomedicines-13-01898-f002:**
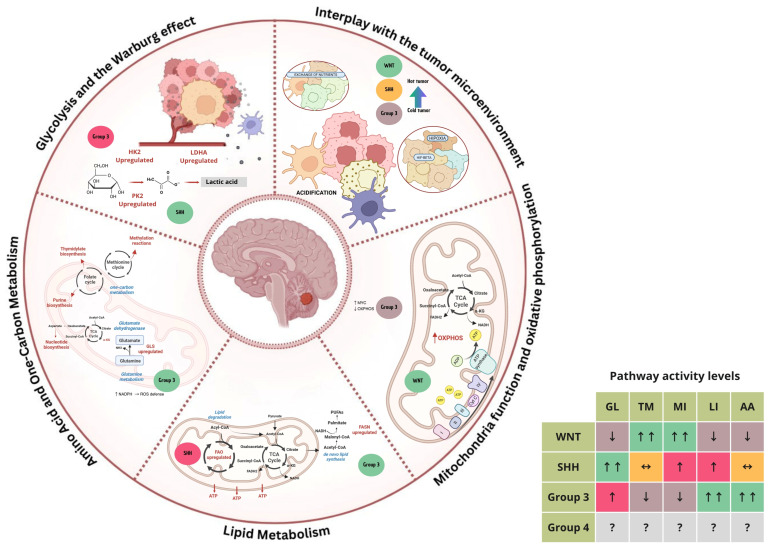
Metabolic Mechanisms in Medulloblastoma. Schematic overview of major metabolic pathways implicated in medulloblastoma, illustrating subtype-specific activity patterns in glycolysis and the Warburg effect, amino acid and one-carbon metabolism, lipid metabolism, mitochondrial function and oxidative phosphorylation (OXPHOS), and interactions with the tumor microenvironment. Distinct pathway alterations are observed among molecular subgroups. Group 3 shows upregulation of glycolytic enzymes (HK2, LDHA, PKM2), enhanced glutamine metabolism, and fatty acid oxidation (FAO), promoting energy production and redox imbalance. SHH-subtype tumors exhibit lipid biosynthesis via FASN (fatty acid synthase) and altered mitochondrial metabolism. The WNT subgroup demonstrates lower metabolic activity overall, while Group 4 remains poorly characterized. The heatmap (bottom right) summarizes relative pathway activity levels across subgroups. Symbol legend: ↑↑ = High activity; ↑ = Moderate activity; ↔ = Balanced or basal activity; ↓ = Low or repressed activity; ? = Poorly characterized. Abbreviations: GL, glycolysis; TM, tumor microenvironment; MI, mitochondrial function and OXPHOS; LI, lipid metabolism; AA, amino acid and one-carbon metabolism; HK2, hexokinase 2; LDHA, lactate dehydrogenase A; PKM2, pyruvate kinase M2; FASN, fatty acid synthase; TCA, tricarboxylic acid cycle; FAO, fatty acid oxidation; GLS, glutaminase; NADH, nicotinamide adenine dinucleotide (reduced form); ROS, reactive oxygen species; PUFA, polyunsaturated fatty acid; OXPHOS, oxidative phosphorylation.

**Figure 3 biomedicines-13-01898-f003:**
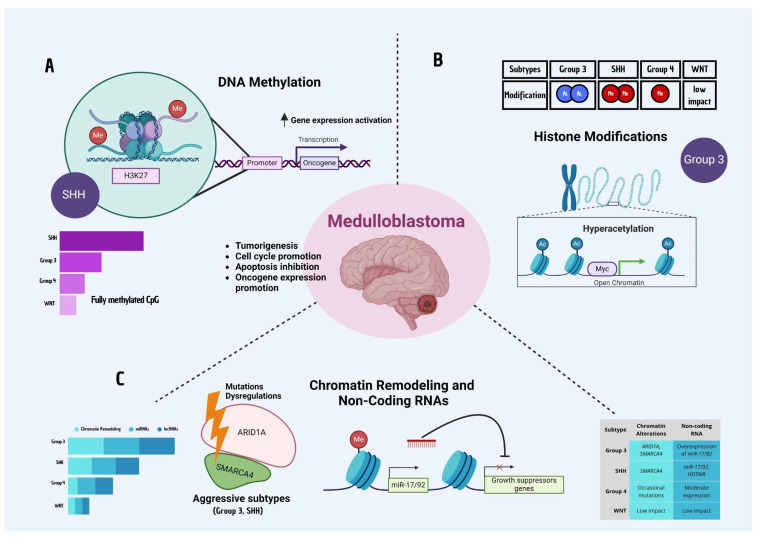
Epigenetic and transcriptional dysregulation in Medulloblastoma. (**A**) DNA Methylation. Schematic representation of DNA methylation at CpG islands, particularly at H3K27, and its effect on gene expression activation in SHH-subtype medulloblastoma. The bar graph shows relative levels of fully methylated CpG sites across subtypes, with the highest levels in SHH and Group 3. DNA methylation contributes to tumorigenesis by promoting oncogene expression, inhibiting apoptosis, and stimulating cell cycle progression. (**B**) Histone Modifications. Group 3 medulloblastoma exhibits histone hyperacetylation, particularly at MYC target loci, associated with open chromatin and transcriptional activation. The upper table summarizes predominant histone modifications by subtype: Group 3 displays acetylation (Ac) and SHH, and Group 4 shows methylation (Me), while WNT exhibits minimal histone modification activity. (**C**) Chromatin Remodeling and Non-Coding RNAs. Aggressive subtypes such as Group 3 and SHH harbor mutations in chromatin remodelers (ARID1A, SMARCA4), along with dysregulation of non-coding RNAs (ncRNAs), including overexpression of oncogenic microRNAs (miRNAs) such as miR-17/92 and long non-coding RNAs (lncRNAs) like HOTAIR. The graph depicts the relative contribution of chromatin remodeling, miRNAs, and lncRNAs across subtypes. The summary table highlights the most relevant epigenetic and ncRNA alterations by molecular group. Abbreviations: CpG, cytosine-phosphate-guanine; H3K27, histone H3 lysine 27; Ac, acetylation; Me, methylation; MYC, MYC proto-oncogene; ARID1A, AT-rich interaction domain 1A; SMARCA4, SWI/SNF related, matrix associated, actin-dependent regulator of chromatin subfamily A member 4; miRNA, microRNA; lncRNA, long non-coding RNA; HOTAIR, HOX transcript antisense RNA; ncRNA, non-coding RNA.

**Figure 4 biomedicines-13-01898-f004:**
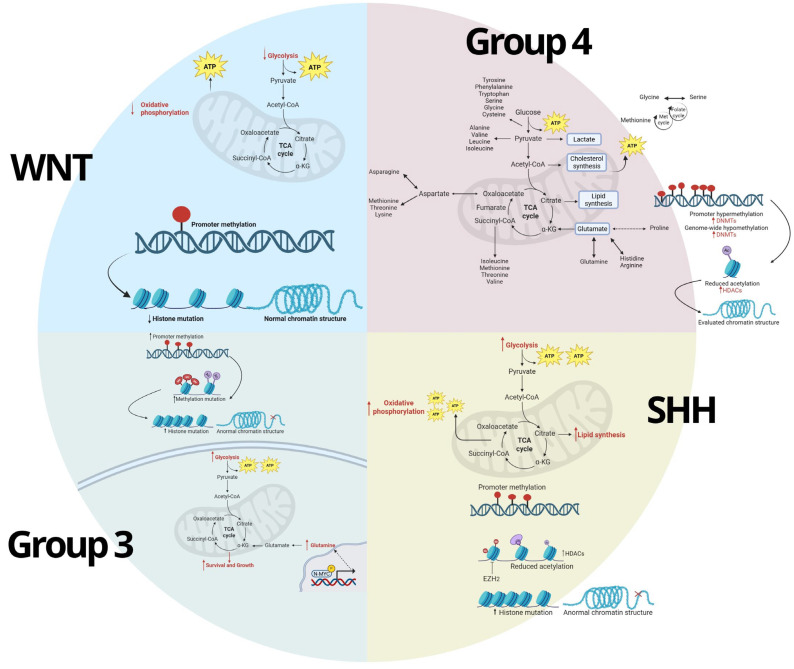
Metabolic and Epigenetic Characteristics of Medulloblastoma. This quadrant diagram illustrates the distinct epigenetic and metabolic hallmarks of the four main medulloblastoma subgroups: WNT, Group 3, SHH (Sonic Hedgehog), and Group 4. WNT subgroup: Characterized by promoter methylation, histone mutations, and preservation of chromatin structure. Energy production relies primarily on oxidative phosphorylation and tricarboxylic acid (TCA) cycle activity. Group 3: Shows promoter hypermethylation, histone and methylation-related mutations, and chromatin disorganization. Metabolically, Group 3 exhibits enhanced glycolysis, glutamine dependence, and altered TCA flux to support tumor growth and survival. SHH subgroup: Displays genome-wide hypomethylation, promoter hypermethylation, and reduced histone acetylation, often mediated by histone deacetylases (HDACs), leading to chromatin compaction. Metabolically, SHH tumors rely on both glycolysis and lipid synthesis for energy and biomass production. Group 4: Although less characterized, this subgroup shows involvement of amino acid metabolism (e.g., glutamate, serine, and methionine), cholesterol and lipid synthesis, and epigenetic alterations such as promoter methylation and chromatin remodeling. Abbreviations: TCA, tricarboxylic acid cycle; ATP, adenosine triphosphate; α-KG, alpha-ketoglutarate; HDAC, histone deacetylase.

**Figure 5 biomedicines-13-01898-f005:**
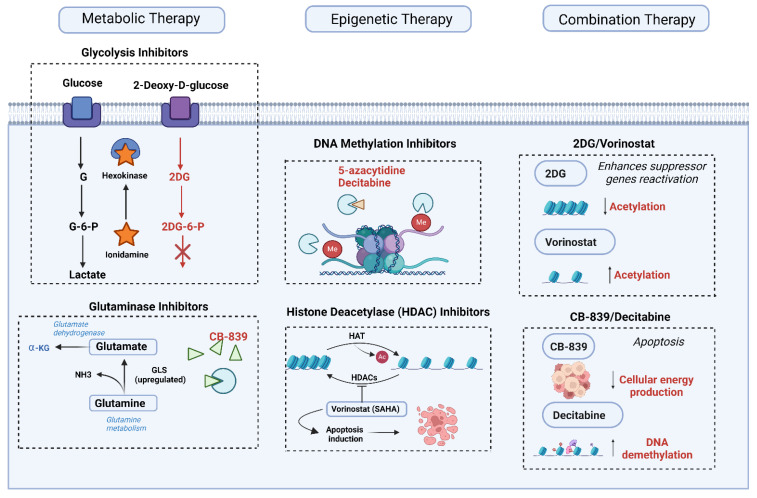
Metabolic and Epigenetic Therapies in Cancer. This schematic summarizes current experimental approaches to inhibit tumor progression in medulloblastoma by targeting altered metabolic and epigenetic pathways. Therapeutic strategies are categorized into metabolic therapy, epigenetic therapy, and combination therapy. Metabolic therapy: Glycolysis inhibitors such as 2-deoxy-D-glucose (2DG) and ionidamine block hexokinase activity, disrupting glucose utilization and ATP production. Glutaminase inhibitors like CB-839 suppress glutamine conversion to glutamate by inhibiting glutaminase (GLS), thereby reducing α-ketoglutarate (α-KG) availability for the TCA cycle. Epigenetic therapy: DNA methylation inhibitors, including 5-azacytidine and decitabine, inhibit DNA methyltransferases, leading to reactivation of silenced tumor suppressor genes. Histone deacetylase (HDAC) inhibitors such as vorinostat (suberoylanilide hydroxamic acid, SAHA) restore acetylation levels by inhibiting HDAC activity, promoting chromatin relaxation and apoptosis through modulation of histone acetylation. Combination therapy: Co-treatment strategies that combine metabolic and epigenetic inhibitors show synergistic effects. 2DG combined with vorinostat enhances tumor suppressor gene reactivation by modulating acetylation. CB-839 combined with decitabine simultaneously impairs energy metabolism and promotes DNA demethylation, increasing tumor cell apoptosis. Abbreviations: 2DG, 2-deoxy-D-glucose; G-6-P, glucose-6-phosphate; GLS, glutaminase; α-KG, alpha-ketoglutarate; HDAC, histone deacetylase; SAHA, suberoylanilide hydroxamic acid; HAT, histone acetyltransferase; Ac, acetylation; ATP, adenosine triphosphate; TCA, tricarboxylic acid cycle.

**Table 1 biomedicines-13-01898-t001:** Comparative Summary of Metabolic and Epigenetic Alterations Across Medulloblastoma Subtypes.

Subtype	Key Metabolic Features	Key Epigenetic Features	Therapeutic Targets (Examples)
*WNT*	Minimal metabolic reprogramming; low glycolytic and OXPHOS activity [7]	Stable DNA methylation and histone profile [10,26]	Limited metabolic targeting; possible use of HDAC inhibitors [26]
*SHH*	Elevated glycolysis, fatty acid synthesis, and mitochondrial OXPHOS [6,19]	EZH2 and HDAC activity; SHH-specific DNA methylation [27,28]	SMO inhibitors (e.g., vismodegib) [19], EZH2 inhibitors [28], HDACi [27]
*Group 3*	High glycolytic flux, glutamine dependency; MYC-driven metabolism [6,24]	Widespread histone acetylation/methylation; BRD4 dependence [29,30]	Glycolysis inhibitors [32], GLS inhibitors [8], BET inhibitors [30]
*Group 4*	Amino acid metabolism (e.g., BCAT1), mitochondrial respiration; SLC transporter upregulation [6,25]	Distinct DNA methylation profiles; HDAC sensitivity [26,31]	Targeting amino acid metabolism or mitochondrial function [25]; HDACi [31]

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
