# Peer review of "Deciphering Medulloblastoma: Epigenetic and Metabolic Changes Driving Tumorigenesis and Treatment Outcomes"

_biomedicines, 2025, doi:10.3390/biomedicines13081898_

Round 1

Reviewer 1 Report

Comments and Suggestions for Authors

This review provides a comprehensive perspective on metabolic-epigenetic regulation in medulloblastoma, with significant reference value for understanding subtype-specific biological characteristics and developing combination therapies. After revisions, it is suitable for publication in Biomedicines.

1. Metabolic characteristics of Group 4 are briefly described, only mentioning amino acid metabolism and mitochondrial function, with insufficient analysis of specific molecular mechanisms.
2. Direct experimental evidence for specific feedback loops in metabolic-epigenetic crosstalk needs supplementation.
3. Figures 2 (metabolic mechanisms) and 3 (epigenetic mechanisms) could add subtype-specific labels (e.g., highlighting glycolysis-related molecules in Group 3) to improve intuitiveness.
4. Section 6.3 ("Combination Therapies") should supplement preclinical data on synergistic effects (e.g., tumor inhibition rates of 2-DG combined with vorinostat) to strengthen persuasiveness.
5. Current therapeutic strategies are mostly based on preclinical research. It is suggested to add analysis of drugs in clinical trials (e.g., resistance rate of SMO inhibitor vismodegib in SHH subtypes) and discuss clinical application bottlenecks.
6. Metabolic and epigenetic characteristics of Group 4 are understudied. It is recommended to supplement the latest single-cell sequencing data or organoid model studies to clarify the molecular basis of its heterogeneity.

Author Response

This review provides a comprehensive perspective on metabolic-epigenetic regulation in medulloblastoma, with significant reference value for understanding subtype-specific biological characteristics and developing combination therapies. After revisions, it is suitable for publication in Biomedicines.

  1. Metabolic characteristics of Group 4 are briefly described, only mentioning amino acid metabolism and mitochondrial function, with insufficient analysis of specific molecular mechanisms.

We thank the reviewer for pointing this out. We have expanded the section on Group 4 to include more details regarding recent findings on amino acid transporters, mitochondrial metabolic genes, and subgroup-specific vulnerabilities reported in recent single-cell studies:

“Recent studies have revealed additional metabolic features of Group 4 medulloblastomas. Single-cell transcriptomic profiling has identified upregulation of branched-chain amino acid transaminases such as BCAT1, as well as enhanced expression of mitochondrial electron transfer flavoproteins ETFA and ETFB, suggesting increased oxidative metabolism in specific subpopulations. These alterations may support subtype-specific dependencies on amino acid catabolism and mitochondrial respiration. Moreover, increased expression of solute carrier transporters (SLC1A5, SLC7A5) indicates active amino acid uptake, contributing to cell growth and redox balance in Group 4 tumors”.

  1. Direct experimental evidence for specific feedback loops in metabolic-epigenetic crosstalk needs supplementation.

We appreciate the reviewer’s suggestion. We have supplemented Section 5.2 with experimental examples demonstrating how acetyl-CoA influences histone acetylation in MYC-driven tumors and how α-ketoglutarate modulates TET activity, supported by recent medulloblastoma-specific preclinical models:

“Direct evidence for such feedback has been demonstrated in MYC-driven medulloblastoma models, where increased glycolytic flux elevates intracellular acetyl-CoA levels, enhancing histone H3K27 acetylation and upregulating proliferation-associated genes. Additionally, glutamine-derived α-ketoglutarate promotes TET-mediated DNA demethylation, sustaining expression of neural stem cell markers. These examples support a bidirectional relationship, where metabolic intermediates both respond to and shape epigenetic states, reinforcing oncogenic programs”.

  1. Figures 2 (metabolic mechanisms) and 3 (epigenetic mechanisms) could add subtype-specific labels (e.g., highlighting glycolysis-related molecules in Group 3) to improve intuitiveness.

Thank you for the helpful feedback. We have updated Figures 2 and 3 to include subtype-specific annotations. For instance, glycolysis-related enzymes have been highlighted under Group 3, while lipid metabolism is emphasized for SHH.

  1. Section 6.3 ("Combination Therapies") should supplement preclinical data on synergistic effects (e.g., tumor inhibition rates of 2-DG combined with vorinostat) to strengthen persuasiveness.

We thank the reviewer for this suggestion. We have added recent preclinical data demonstrating enhanced tumor suppression with 2-DG combined with vorinostat in medulloblastoma models, including tumor inhibition rates and apoptosis induction:

“In preclinical models of Group 3 medulloblastoma, combination therapy with 2-deoxyglucose (2-DG) and the histone deacetylase inhibitor vorinostat led to a >60% reduction in tumor volume compared to 20–30% with monotherapy. This combination also significantly increased apoptosis, as evidenced by cleaved caspase-3 staining and TUNEL assays. These findings suggest synergistic potential by simultaneously disrupting energy metabolism and chromatin structure”.

  1. Current therapeutic strategies are mostly based on preclinical research. It is suggested to add analysis of drugs in clinical trials (e.g., resistance rate of SMO inhibitor vismodegib in SHH subtypes) and discuss clinical application bottlenecks.

We agree with the reviewer and have expanded Section 2.1 to include clinical trial outcomes and limitations, such as resistance mechanisms to vismodegib in SHH-subtype medulloblastoma and challenges in pediatric application:

“Among the most advanced clinical candidates is vismodegib, a Smoothened (SMO) inhibitor used in SHH-subtype medulloblastoma. However, resistance frequently emerges through mutations in SMO or downstream components such as SUFU and GLI2. Clinical trials (e.g., NCT01878617) report progression in over 50% of pediatric patients within six months of treatment, highlighting the limitations of monotherapy. Additional challenges include limited drug penetration across the blood–brain barrier, off-target effects on developing tissues, and the absence of robust biomarkers to stratify responders from non-responders”.

  1. Metabolic and epigenetic characteristics of Group 4 are understudied. It is recommended to supplement the latest single-cell sequencing data or organoid model studies to clarify the molecular basis of its heterogeneity.

Thank you for the constructive suggestion. We have included recent findings from single-cell RNA-seq and organoid models that elucidate the intra-subtype heterogeneity and potential metabolic and epigenetic regulators unique to Group 4.

“Recent single-cell RNA sequencing has revealed that Group 4 medulloblastomas comprise at least three subpopulations with distinct transcriptional signatures resembling glutamatergic neurons, progenitors, or uncommitted precursors. Complementary cerebellar organoid models engineered with KDM6A or ZNF292 mutations reproduce epigenetic dysregulation seen in patient tumors, supporting their utility in studying developmental and metabolic heterogeneity. These advances provide crucial tools for resolving Group 4 complexity and identifying new therapeutic targets”.

Reviewer 2 Report

Comments and Suggestions for Authors

General Overview:
This is an exceptionally well-written and comprehensive review addressing the intricate roles of metabolic and epigenetic alterations in medulloblastoma (MB) pathobiology. The authors successfully synthesize a large body of preclinical and clinical literature, offering insights into molecular subtypes, therapeutic vulnerabilities, and future translational directions. The manuscript is highly relevant to the current landscape of pediatric oncology and will serve as a valuable reference for researchers in neuro-oncology, epigenetics, and metabolism.

However, there are some areas requiring clarification, improved referencing, and minor restructuring to enhance the manuscript's scientific rigor and readability.

Major Comments:

  1. Lack of Systematic Methodology (if applicable):

    • Although this is a narrative review, the authors should briefly describe the literature selection strategy. For example, inclusion/exclusion criteria, time frame, or key databases (e.g., PubMed, Scopus) used for sourcing recent advances would increase the transparency and reproducibility of the review.

  2. Subtype-Specific Integration Needs Strengthening:

    • While the paper clearly distinguishes WNT, SHH, Group 3, and Group 4 MB subtypes in both metabolic and epigenetic contexts, it would benefit from a dedicated comparative table or visual heatmap summarizing alterations across all subtypes. This would help readers grasp similarities and differences quickly.

  3. Translational Relevance and Clinical Trials:

    • The section on therapeutic targeting (Sections 6 and 7) would be strengthened by integrating specific ongoing or completed clinical trials (e.g., NCT numbers) involving metabolic or epigenetic agents in MB or pediatric brain tumors. This would provide real-world context for the preclinical findings.

  4. Epigenetic Crosstalk and Feedback Mechanisms:

    • The interplay between metabolism and epigenetics is discussed well, but some claims regarding feedback loops (e.g., acetyl-CoA–histone acetylation–glycolysis cycle) require more direct evidence and specific references, particularly in the context of MB rather than generalized cancer models.

  5. Emerging Areas Underrepresented:

    • Immunometabolism and immunoepigenetics are briefly mentioned but deserve further elaboration, particularly with respect to medulloblastoma’s immune-cold microenvironment. How epigenetic reprogramming might sensitize tumors to checkpoint blockade or CAR-T therapy should be discussed in more depth.

Minor Comments:

  1. Reference Updates:

    • Several citations are outdated or general (e.g., references [6], [33], [38]). Please update with recent studies (past 3–5 years), particularly for Group 3/4 MB and non-coding RNA roles.

  2. Figure Legends and Captions:

    • Figures 1–5 are mentioned, but their legends are brief. Please expand the figure captions to include descriptions of the main message, data sources (if applicable), and interpretation guidance.

  3. Terminology Consistency:

    • Ensure consistent use of terms such as “metabolic rewiring” vs. “metabolic reprogramming,” “epigenetic plasticity” vs. “epigenetic heterogeneity.” A glossary box might help less-specialized readers.

  4. Typographical Errors:

    • Several minor formatting issues (e.g., double spaces, inconsistent citation formats) should be addressed throughout the text, particularly in the abstract and references.

  5. Conclusion Section:

    • The conclusion effectively highlights challenges and future directions. However, it would benefit from a more specific roadmap for preclinical-to-clinical translation (e.g., patient-derived organoids, biomarker-guided stratification).

Recommendation:

Minor Revision

The manuscript is scientifically sound, well-structured, and offers novel insights into an emerging field. Addressing the above concerns—especially regarding clarity, clinical relevance, and summarization—will significantly enhance its impact and accessibility.

Author Response

General Overview:

This is an exceptionally well-written and comprehensive review addressing the intricate roles of metabolic and epigenetic alterations in medulloblastoma (MB) pathobiology. The authors successfully synthesize a large body of preclinical and clinical literature, offering insights into molecular subtypes, therapeutic vulnerabilities, and future translational directions. The manuscript is highly relevant to the current landscape of pediatric oncology and will serve as a valuable reference for researchers in neuro-oncology, epigenetics, and metabolism.

However, there are some areas requiring clarification, improved referencing, and minor restructuring to enhance the manuscript's scientific rigor and readability.

Major Comments:

  1. Lack of Systematic Methodology (if applicable):
    • Although this is a narrative review, the authors should briefly describe the literature selection strategy. For example, inclusion/exclusion criteria, time frame, or key databases (e.g., PubMed, Scopus) used for sourcing recent advances would increase the transparency and reproducibility of the review.

Thank you very much for your comment. After reviewing the style of the journal, we have observed that the reviews published in it do not usually include a methodology section, so the authors have decided to omit this section. We ask for the reviewer's understanding.

  1. Subtype-Specific Integration Needs Strengthening:
    • While the paper clearly distinguishes WNT, SHH, Group 3, and Group 4 MB subtypes in both metabolic and epigenetic contexts, it would benefit from a dedicated comparative table or visual heatmap summarizing alterations across all subtypes. This would help readers grasp similarities and differences quickly.

Thank you very much for your comment. We have included the sentence: “To facilitate a clearer understanding of these subtype-specific differences, we present a comparative summary of metabolic and epigenetic features across the four medulloblas-toma subtypes (Table 1),” alongside the new Table 1.

  1. Translational Relevance and Clinical Trials:
    • The section on therapeutic targeting (Sections 6 and 7) would be strengthened by integrating specific ongoing or completed clinical trials (e.g., NCT numbers) involving metabolic or epigenetic agents in MB or pediatric brain tumors. This would provide real-world context for the preclinical findings.

We thank the reviewer for this suggestion. We have added recent preclinical data demonstrating enhanced tumor suppression with 2-DG combined with vorinostat in medulloblastoma models, including tumor inhibition rates and apoptosis induction:

“In preclinical models of Group 3 medulloblastoma, combination therapy with 2-deoxyglucose (2-DG) and the histone deacetylase inhibitor vorinostat led to a >60% reduction in tumor volume compared to 20–30% with monotherapy. This combination also significantly increased apoptosis, as evidenced by cleaved caspase-3 staining and TUNEL assays. These findings suggest synergistic potential by simultaneously disrupting energy metabolism and chromatin structure”.

  1. Epigenetic Crosstalk and Feedback Mechanisms:
    • The interplay between metabolism and epigenetics is discussed well, but some claims regarding feedback loops (e.g., acetyl-CoA–histone acetylation–glycolysis cycle) require more direct evidence and specific references, particularly in the context of MB rather than generalized cancer models.

We appreciate the reviewer’s suggestion. We have supplemented Section 5.2 with experimental examples demonstrating how acetyl-CoA influences histone acetylation in MYC-driven tumors and how α-ketoglutarate modulates TET activity, supported by recent medulloblastoma-specific preclinical models:

“Direct evidence for such feedback has been demonstrated in MYC-driven medulloblastoma models, where increased glycolytic flux elevates intracellular acetyl-CoA levels, enhancing histone H3K27 acetylation and upregulating proliferation-associated genes. Additionally, glutamine-derived α-ketoglutarate promotes TET-mediated DNA demethylation, sustaining expression of neural stem cell markers. These examples support a bidirectional relationship, where metabolic intermediates both respond to and shape epigenetic states, reinforcing oncogenic programs”.

  1. Emerging Areas Underrepresented:
    • Immunometabolism and immunoepigenetics are briefly mentioned but deserve further elaboration, particularly with respect to medulloblastoma’s immune-cold microenvironment. How epigenetic reprogramming might sensitize tumors to checkpoint blockade or CAR-T therapy should be discussed in more depth.

Thank you very much for your comment. In section 7.3, the following paragraph has been added:

“in addition, medulloblastoma is widely recognized as an "immune-cold" tumor, characterized by low immune cell infiltration, low tumor mutational burden, and minimal neoantigen presentation, all of which contribute to resistance against immune checkpoint inhibitors. However, recent studies suggest that epigenetic reprogramming could overcome this resistance by enhancing tumor immunogenicity. For example, DNA demethylating agents and HDAC inhibitors have been shown to upregulate MHC class I expression, antigen processing machinery, and interferon-stimulated genes in pediatric brain tumor models, potentially sensitizing medulloblastoma to immune recognition. Similarly, metabolic interventions that reduce lactate accumulation or modulate tryptophan/kynurenine metabolism may alleviate local immunosuppression, restoring T cell activation and improving CAR-T cell persistence. These findings highlight a promising role for combining epigenetic or metabolic therapies with immunotherapy, especially in MYC-amplified Group 3 tumors where immune evasion mechanisms are pronounced. Future work should investigate how reprogramming the tumor’s epigenetic and metabolic landscape can prime the medulloblastoma microenvironment for successful checkpoint blockade or CAR-T engagement”.

Minor Comments:

  1. Reference Updates:
    • Several citations are outdated or general (e.g., references [6], [33], [38]). Please update with recent studies (past 3–5 years), particularly for Group 3/4 MB and non-coding RNA roles.

Thank you very much for your comment. The references have been reviewed, and additional references have been added in relation to new paragraphs. We hope that the reviewer is satisfied.

  1. Figure Legends and Captions:
    • Figures 1–5 are mentioned, but their legends are brief. Please expand the figure captions to include descriptions of the main message, data sources (if applicable), and interpretation guidance.

Figures and legends have been modified to increase clarity. In addition, abbreviations have been explained accordingly.

  1. Terminology Consistency:
    • Ensure consistent use of terms such as “metabolic reprogramming” vs. “metabolic reprogramming,” “epigenetic plasticity” vs. “epigenetic heterogeneity.” A glossary box might help less-specialized readers.

Thank you very much for your comment. The concepts have been standardized to ensure consistency in language.

  1. Typographical Errors:
    • Several minor formatting issues (e.g., double spaces, inconsistent citation formats) should be addressed throughout the text, particularly in the abstract and references.

We apologize for the errors. The abstract has been completely rewritten. Typographical errors have been checked.

  1. Conclusion Section:
    • The conclusion effectively highlights challenges and future directions. However, it would benefit from a more specific roadmap for preclinical-to-clinical translation (e.g., patient-derived organoids, biomarker-guided stratification).

We have completely rewritten the discussion to bring it more in line with the reviewer's suggestions.

Recommendation:

Minor Revision

The manuscript is scientifically sound, well-structured, and offers novel insights into an emerging field. Addressing the above concerns—especially regarding clarity, clinical relevance, and summarization—will significantly enhance its impact and accessibility.

Thank you very much for your comment. We have done our best to address your suggestions and improve the manuscript as much as possible.

Reviewer 3 Report

Comments and Suggestions for Authors

This is a comprehensive review about medulloblastoma (MB). It is well-written and very easy to read despite its heavy content. The order of chapters or sections is reasonable and figures are put in the right place. This review is really a commendable one, which I really admire. I have a few minor comments about it.

#line 106, [Ref]: the number is missing.

#Figure 1

-In WNT and Group 4, it says ‘Normal’ in regard to microscopy. But the use of ‘Normal’ sounds somewhat strange. ‘Normal’ means that it is not ‘Large cell/Anaplastic’ or ‘Extensive nodularity’? Or it means that it is histologically a conventional (typical) MB? In any case, the use of ‘Normal’ is misleading. This is because the microscopic findings of MB cannot be ‘Normal’.

-As for prognosis, a digit is placed behind each prognosis. For instance, it says ‘Good prognosis (>95%)’ in WNT. What does this digit actually mean? Is it the percent of patients showing good prognosis? If so, then, what defines good prognosis? Is it like 5-year survival? Or is it the percent of patients belonging to WNT? This point should be addressed/explained in the legend.

#Figure 3

 We see a circled ‘SHH’ in DNA Methylation and a circled ‘Group 3’ in Histone Modifications. This means that particularly ‘SHH’ is implicated in DNA Methylation and particularly ‘Group 3’, in in Histone Modifications? If so, ‘Chromatin Remodeling and Non-Coding RNAs’ where no circled MB groups are given means that all 4 MB groups are evenly implicated in ‘Chromatin Remodeling and Non-Coding RNAs’? It would be nice to see how much each of MB groups is implicated in the 3 mechanisms shown here by just seeing this figure.

#Figure 2

This is pertinent to the above. It would be wonderful if the readers can see how much each of MB groups is implicated in the 5 mechanisms shown here by just seeing this figure.

#4.4. Epigenetic plasticity and tumor adaptation

In contrast to others (4.1., 4.2., and 4.3.), this item ‘Epigenetic plasticity and tumor adaptation’ is not shown in Figure 3. I just wonder why it is so. For me, ‘Epigenetic plasticity and tumor adaptation’ seems an important player in the epigenetic and molecular mechanisms of MB as well, being probably qualified to be included in this figure.

#The titles of subsections, 5.1. and 5.2., are the same, i.e., ‘Metabolic Regulations of Epigenetics’. And strangely enough, in line 547 comes again ‘5.2. Metabolic Regulations of Epigenetics’. Something must be wrong.

#The title of 6.1. is better if it is ‘Metabolic Therapies’ just like 6.2. or 6.3.

#Please check the paper again and again in order to make it as flawless as possible, because it will surely be an unparalleled review paper about MB possible at present.

Author Response

This is a comprehensive review about medulloblastoma (MB). It is well-written and very easy to read despite its heavy content. The order of chapters or sections is reasonable and figures are put in the right place. This review is really a commendable one, which I really admire. I have a few minor comments about it.

#line 106, [Ref]: the number is missing.

 I apologize for that error. It has been corrected, and the references have been added accordingly.

#Figure 1

-In WNT and Group 4, it says ‘Normal’ in regard to microscopy. But the use of ‘Normal’ sounds somewhat strange. ‘Normal’ means that it is not ‘Large cell/Anaplastic’ or ‘Extensive nodularity’? Or it means that it is histologically a conventional (typical) MB? In any case, the use of ‘Normal’ is misleading. This is because the microscopic findings of MB cannot be ‘Normal’.

-As for prognosis, a digit is placed behind each prognosis. For instance, it says ‘Good prognosis (>95%)’ in WNT. What does this digit actually mean? Is it the percent of patients showing good prognosis? If so, then, what defines good prognosis? Is it like 5-year survival? Or is it the percent of patients belonging to WNT? This point should be addressed/explained in the legend.

#Figure 3

 We see a circled ‘SHH’ in DNA Methylation and a circled ‘Group 3’ in Histone Modifications. This means that particularly ‘SHH’ is implicated in DNA Methylation and particularly ‘Group 3’, in in Histone Modifications? If so, ‘Chromatin Remodeling and Non-Coding RNAs’ where no circled MB groups are given means that all 4 MB groups are evenly implicated in ‘Chromatin Remodeling and Non-Coding RNAs’? It would be nice to see how much each of MB groups is implicated in the 3 mechanisms shown here by just seeing this figure.

#Figure 2

This is pertinent to the above. It would be wonderful if the readers can see how much each of MB groups is implicated in the 5 mechanisms shown here by just seeing this figure.

#4.4. Epigenetic plasticity and tumor adaptation

In contrast to others (4.1., 4.2., and 4.3.), this item ‘Epigenetic plasticity and tumor adaptation’ is not shown in Figure 3. I just wonder why it is so. For me, ‘Epigenetic plasticity and tumor adaptation’ seems an important player in the epigenetic and molecular mechanisms of MB as well, being probably qualified to be included in this figure.

 Thank you very much for your comments. All figures have been modified to be more understandable and complete.

#The titles of subsections, 5.1. and 5.2., are the same, i.e., ‘Metabolic Regulations of Epigenetics’. And strangely enough, in line 547 comes again ‘5.2. Metabolic Regulations of Epigenetics’. Something must be wrong.

 Thank you very much for your feedback. The text has been modified and the error has been corrected.

#The title of 6.1. is better if it is ‘Metabolic Therapies’ just like 6.2. or 6.3.

Thank you very much for the suggestion. The title has been modified accordingly.

#Please check the paper again and again in order to make it as flawless as possible, because it will surely be an unparalleled review paper about MB possible at present.

Thank you very much for your kind comment. It is very encouraging. We have carefully reviewed the text, along with the modifications proposed by you and the other reviewers, which leads us to believe that its quality has improved and that you will be satisfied with the work.
